# Polygyny is linked to accelerated birdsong evolution but not to larger song repertoires

Kate T. Snyder[1] & Nicole Creanza[1]

Non-monogamous mating behaviors including polygyny or extra-pair paternity are theorized to amplify sexual selection, since some males attract multiple mates or copulate with paired females. In several well-studied songbird species, females prefer more complex songs and larger repertoires; thus, non-monogamous mating behaviors are predicted to accelerate song evolution, particularly toward increased complexity. However, studies within songbird clades have yielded mixed results, and the effect of non-monogamy on song evolution remains unclear. Here, we construct a large-scale database synthesizing mating system, extra-pair paternity, and song information and perform comparative analyses alongside songbird genetic phylogenies. Our results suggest that polygyny drives faster evolution of syllable repertoire size (measured as average number of unique syllables), but this rapid evolution does not produce larger repertoires in polygynous species. Instead, both large and small syllable repertoires quickly evolve toward moderate sizes in polygynous lineages. Contrary to expectation, high rates of extra-pair paternity coincide with smaller repertoires.

[1] Department of Biological Sciences, Vanderbilt University, 465 21st Avenue South, Nashville, TN 37240, USA. Correspondence and requests for materials should be addressed to N.C. (email: nicole.creanza@vanderbilt.edu)

Polygyny, a social mating system in which one male can be mated to several females simultaneously, has evolved numerous times in birds[1,2]. Since polygynous male birds have the potential to raise clutches of offspring with multiple females, it is expected that there will be substantially higher variance in the fitness of individual males in polygynous species compared to monogamous species in which males have only one social mate at a time[3,4]. In other words, a polygynous male could multiply his potential number of offspring by the number of mates he can attract, leading to differences in reproductive success that could be much larger than the fitness differences of only a few percent that have been shown to shift evolutionary dynamics in a population (e.g., refs. [5,6]). A polygynous mating strategy also leaves more males unpaired than monogamy; these males could have zero reproductive success unless they were successful in extra-pair fertilizations (EPF)[7]. Since the stakes are higher to successfully attract one or more mates, the high variance in reproductive success predicted for polygynous species could amplify the role of sexual selection compared to monogamous species.

Extra-pair paternity (EPP) is common in Oscine species and, like polygyny, may affect the variance of reproductive success in males in a population[8,9]. In both monogamous and polygynous species where EPP is common, females will often form a social bond with a single male partner but also copulate with other males[10]. Seeking extra-pair fertilizations could act as a bet-hedging strategy that, in theory, could increase a female's indirect reproductive fitness, for example by increasing the genetic diversity or the fitness of her offspring[11,12]. Since sexual selection can occur when choosing these extra-pair mates[13,14], males with less attractive secondary characteristics may be doubly penalized; once in obtaining a social mate, and again in attaining copulations with already-mated females. Thus, EPP could potentially increase variance in reproductive fitness (e.g., ref. [15], but see ref. [16]). On the other hand, EPP could putatively decrease the variance in fitness in a population, since socially paired males might invest resources in non-genetically-related offspring due to EPP, and even unpaired males would have the opportunity to sire offspring via EPP (e.g., ref. [17]). In this context, it is unclear what kind of association we would expect to find between EPP and learned song. On one hand, females could potentially diversify their mates by exhibiting a different set of song preferences when they seek out extra-pair copulations from when they choose social mates, thereby placing context-dependent evolutionary pressure on song. Alternatively, high EPP could cause more frequent opportunities for females to act upon the same preferences displayed for social mates, which would hypothetically drive song complexity higher.

For songbirds (class: Aves, order: Passeriformes, suborder: Oscine), song is an important and nearly ubiquitous behavior that has functions in mate choice, intrasexual competition, and mediating other social interactions[18–20]. Birdsong could be a particularly salient target for sexual selection since, as a learned behavior, it has the potential for rapid change over generations and large variation within a population. Indeed, songs of male songbirds have been demonstrated to influence mate choice in several species; female songbirds across multiple passerine families have been shown to choose mates with larger syllable repertoires, more syllables per song, longer songs, and larger song repertoires[21–23]. For example, in the great reed warbler, a polygynous species, females preferred larger repertoires both when selecting a social mate and when seeking extra-pair copulation and fertilization[24]. However, counter-examples have also been reported; for instance, female collared flycatchers appeared to prefer males that had smaller repertoires[25].

These findings suggest a possible link between the mating system of a species and its song evolution. In particular, a hypothesis has been repeatedly proposed: the intensified sexual selection in polygynous mating systems should drive the evolution of increased song complexity[26,27]. Several papers have investigated this hypothesis that mating system and song evolution are not independent, with mixed results[28] (Table 1). For example, a study of nine species of North American wrens found that the polygynous species had larger song repertoires[29]. In a meta-analysis of 142 species across several families that did not account for shared ancestry beyond superfamily classification, Read and Weary[30] found that polygynous species have higher numbers of syllables per song. In contrast, a study of 21 New World blackbird species found no correlation between song and syllable repertoire sizes and mating system[27]. Further, two studies showed that the two polygynous species of *Acrocephalus* warblers had simpler repertoires than the four monogamous species included in the studies[26,31]. Taken together, these results hint at a possible relationship between song and mating system, but it does not appear to consistently follow the prediction that increased sexual selection via non-monogamous mating systems favors more elaborate song repertoires.

EPP has been previously studied in the context of song characteristics, but mostly within the scope of a single species. Some studied species show a positive trend: rates of EPP were higher in males with larger repertoires (great reed warblers[24]) or with increased song diversity (reed bunting[32]). In contrast, there was no observed correlation between song characteristics and EPP in the song sparrow[33] and a negative correlation, with greater EPP associated with smaller repertoires, in the sedge warbler[34]. Further, some studies that tested for a correlation between repertoire size and rates of EPP found other factors to be predictive instead, such as singing earlier (blue tit[35]), more consistently (chestnut-sided warbler[36]), or at higher amplitude (dusky warbler[37]). On a larger scale, EPP was not correlated with song complexity in a study of 65 species in which the authors controlled for shared ancestry but did not include syllable repertoire data[38]. More recently, a study across 78 species indicated that within-song complexity (e.g., syllables per song, song duration) was positively correlated with rates of EPP occurrence, but between-song complexity (e.g., song repertoire, syllable repertoire) was not[39] (Table 1).

Here, we seek to understand the relationship between mating strategies and song across the evolutionary history of songbirds through computational and phylogenetically informed analyses. First, we broaden and deepen the scope of this type of analysis by compiling data from many sources on mating system, EPP, and numerous song characteristics from published literature and curated field guides (see Table 1 for definitions of these traits). Gathering data on multiple song metrics enables us to study features of song that have been traditionally categorized as measurements of song complexity, such as total individual syllable repertoire, average syllables per song, and individual song repertoire, as well as features that are often used to measure song performance, such as average song duration, intersong interval, and computed metrics of song rate and continuity. In contrast to most prior studies, we analyze song evolution in the context of both mating system (monogamy/polygyny) and rate of EPP. We perform these analyses while controlling for shared ancestral history using large-scale avian phylogenies[40] of the Oscine suborder.

Our findings suggest a more complex evolutionary relationship between polygyny and EPP and sexual selection on song characteristics than previously has been hypothesized. We test numerous song parameters in this context; in particular, we highlight results from our analyses of syllable repertoire size, a measure of song

### Table 1 Previous comparisons of song and mating system evolution

| Citation | # Species | Phylogenetic control | Song parameters | Mating parameters | Test | Results |
|---|---|---|---|---|---|---|
| Kroodsma (1977)[29] | 9 | One family: Troglodytidae | Syllable repertoire, Syllables/ song, Song-type repertoire, Duration, Continuity | Monogamy/ Polygyny | Qualitative observation | Polygynous species had longer and more complex songs, spent more bout time singing, and switched songs more rapidly |
| Catchpole (1980)[26] | 6 | One family: Acrocephalidae | Duration, Complexity | Monogamy/ Polygyny | Qualitative observation | Two polygynous species: shorter, simpler, less variable songs |
| Catchpole and McGregor (1985)[49] | 5 | One family: Emberizidae | Song repertoire, Variability within a population | Monogamy/ Polygyny | Qualitative observation | One polygynous species: smaller song repertoire, less variation within populations |
| Irwin (1990)[27] | 17 | One family: Icteridae analyzes more closely-related species first (5 groups: cowbirds, grackles, ageline blackbirds, meadowlarks, orioles/ caciques) | Syllable repertoire, Song repertoire, Versatility | Monogamy/ Polygyny | Rank order | Agelaius blackbirds and cowbirds: versatility associated with monogamy. Orioles/ caciques: syll. rep possibly associated with polygyny. Grackles: versatility associated with polygyny |
| Shutler and Weatherhead (1990)[50] | 56 | One family: Parulinae Some analyses within genera | Syllables/song, Song repertoire, Duration, Song rate, Time singing, Frequency | Monogamy/ Polygyny | Mann–Whitney | Monogamous species had larger syllable repertoires |
| Read and Weary (1992)[30] | 142 | Test within superfamilies: Tyrannoidea, Corvoidea, Fringilloidea, Sylvioidea, Turdoidea | Syllables/song, Song repertoire, Interval, Duration, Song rate, Continuity, Versatility | Monogamy/ Polygyny | Binomial Rank order | Polygyny associated with lower song rates across all species, Sylls/song positively associated with polygyny across all species |
| Garamszegi and Møller (2004)[38] | 65 | Phylogenetic control— generalized least squares models via software Continuous (Pagel, 1997, 1999) | Syllables/song, Song repertoire, Interval, Duration, Song rate, Continuity, Versatility | EPP (Continuous) | Generalized least squares models for continuous variables | No correlation between song characteristics and EPP |
| Soma and Garamszegi (2011)[23] | 26, 24 | None (for these data) | "Complexity" term encompassing syllable repertoire, song repertoire, and song versatility | EPP (3 groups); Monogamy, Fac. Polygyny, Polygyny | Meta-regression analysis | No significant correlation between song complexity and EPP or mating system |
| Hill et al. (2017)[39] | 78 | Phylogenetic control— PGLS analysis | Syllable repertoire, Syllables/ song, Song repertoire, Duration, Versatility, Syll. transitions/song, Within-song complexity | EPP (continuous); Monogamy/ "Polygamy"/ Cooperative | Linear regression | Syllables per song (unique), syllable transitions per song, overall within-song complexity positively correlated with EPP |
| Current study | 890 | Phylogenetic control | Syllable repertoire ($N = 120$), Syllables/song ($N = 178$), Song repertoire ($N = 225$), Interval ($N = 131$), Duration ($N = 241$), Song rate ($N = 126$), Continuity ($N = 126$) | EPP (Low/ High) ($N = 142$); Monogamy/ Polygyny ($N = 764$) | PhylANOVA, Brownie, BayesTraits, PGLS, GLMM (see Methods) | Syllable repertoire and song duration evolve faster in polygynous species; Syllable repertoire is smaller in species with high EPP |

Definitions of song terms tested in this study are provided in Table 2. Some previous studies used different terms to refer to the same behavioral trait; see Methods. Definitions of mating system terms used in this study: Monogamy/Polygyny: social monogamy vs. social polygyny, based on qualitative or quantitative descriptions. If quantitative, populations with <5% males with multiple social mates classified as monogamous. EPP: extra-pair paternity, primarily quantitative based on genetic parentage testing of chicks in a population. Species with <10% offspring in a population sired by male who is not the social mate of the female considered to have a low rate of EPP

complexity that enables comparison across many songbird families. Contrary to the hypothesis, we find that polygynous birds do not have systematically larger syllable repertoires than monogamous birds. However, we do find a significant difference in the rate of song evolution between polygynous and monogamous species: syllable repertoire size seems to have evolved significantly faster in polygynous lineages, but this rapid evolution does not push the syllable repertoire size consistently higher. Instead, we find that the combination of polygyny and very small repertoires or very large repertoires are both evolutionarily unstable; polygyny seems to drive the evolution of more moderate-sized repertoires. Our analyses of EPP and song characteristics also yield results that run counter to expectation: we find that syllable repertoire sizes are significantly larger in species with low EPP, and the combination of low EPP and small syllable repertoires is a rare and seemingly unstable state in evolutionary history.

## Results

**Assembled database.** Here, we compiled a database of mating systems and song characteristics across the songbird lineage from published literature and curated field guides (data in Supplementary Data 1, variable description in Table 1, full data curation protocol in Methods); this database includes 764 species with a mating system classification of either social monogamy or polygyny and 141 species that could be identified as having high or low incidence of EPP. In addition, we catalogued song characteristic data (defined in Table 2) for 352 oscine species: 122 species with syllable repertoire size data, 171 species with syllables per song data, 217 species with song repertoire size data, 127 species with song interval data, 228 species with song duration data, and 122 species with both song interval and duration data, allowing calculation of song rate and continuity. In total, the dataset catalogues two mating strategy traits and seven song traits; 890 species across 79 families have data for one or more of these traits (Table 1). Thirty-nine of these species are suboscines, an outgroup to the oscine songbirds in which learned song has not been observed; thus, they were not used in the analyses testing correlation between mating system and song evolution in this study. We used these species to root our phylogenetic trees and included any with relevant mating system/EPP data in the calculation of the rates of transition between monogamy and polygyny and between low EPP and high EPP, which were used in later analyses. When we found multiple measures for the same species, we included all of them in our database. If an analysis required a single species measure for a song feature, we

**Table 2 Definitions for song characteristics used in this paper**

| Song trait | Definition |
|---|---|
| Syllable repertoire | Mean total number of unique syllables an individual uses across songs |
| Syllables per song | Mean number of unique syllables used per song |
| Song repertoire | Mean total number of unique songs an individual produces |
| Intersong interval | Mean length of time separating songs within a period of consistent singing behavior (unit: seconds) |
| Song duration | Mean length of a song, measured as the length of time of consistent singing or discrete songs between periods of silence; sources may have differed in definition based on the song structure of a studied species (unit: seconds) |
| Song rate | Number of full song cycles produced per minute, computed from song duration and intersong interval values |
| Song continuity | Proportion of total song performance time spent producing song, computed from song duration and intersong interval values |

Here, we note the definitions that we used throughout our analyses. Some previous studies have characterized birdsong in different terms; for example, what we term a "syllable" is also called a strophe, note, element, etc. Further, Read and Weary[30] define "syllable repertoire" as the number of syllables in a single song, whereas we classified those data as "syllables per song" in our database. When we gathered song characteristic data from cited sources, we classified these data according to the definitions given in that source, regardless of the terms used

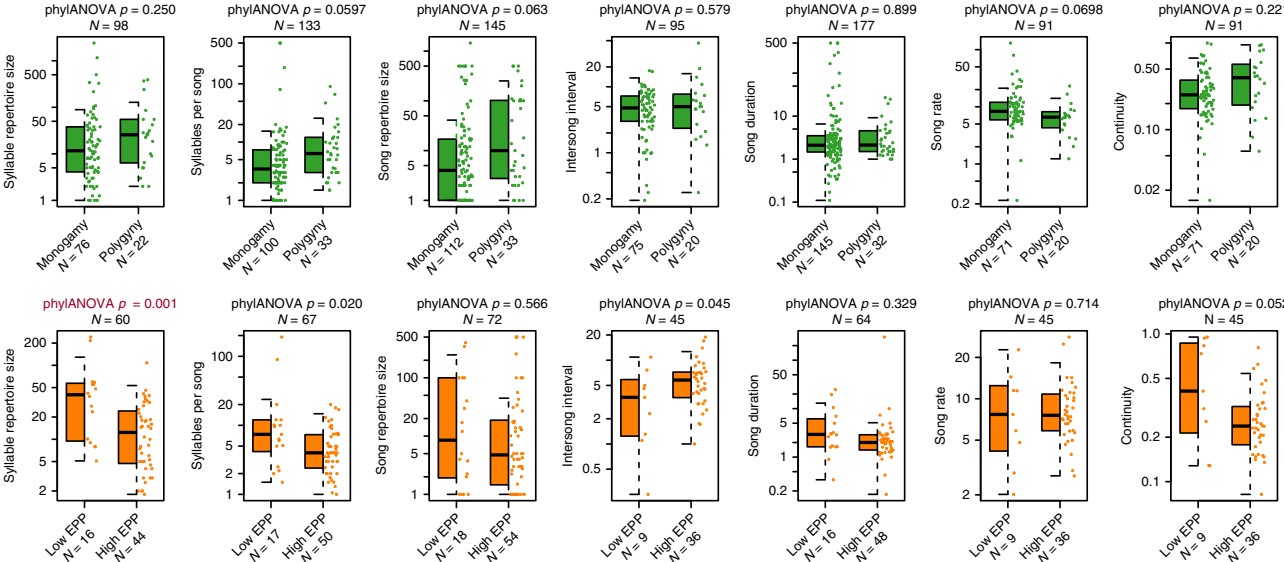

**Fig. 1** Differences in song characteristics for different rates of polygyny and extra-pair paternity. Each plot shows the distribution of song characteristics for species in our database. The top row (green) compares monogamous and polygynous species, with polygyny defined as >5% of males taking multiple mates when quantitative data is available, and the bottom row (orange) compares species with low and high rates of EPP, with a high rate of EPP defined as >10% of offspring in a population being the product of extra-pair fertilization (see Methods for full classification criteria). Box plots indicate the median (black bar) interquartile range (IQR, box) and Q1 − 1.5*IQR, Q3 + 1.5*IQR (whiskers) of each distribution, and scatter plots of the data are shown to the right of each box plot. We compared each pair of distributions phylogenetic ANOVA (phylANOVA) tests, which control for shared ancestry. p-values for these tests are shown above each box plot, with statistically significant results shown in red

performed the analysis on the median value for each species and repeated the analysis with minimum and maximum values. If the analysis could accommodate multiple measures from a species, we included all measures.

**Correlations between song characteristics and mating system.** To test whether mating system and EPP were associated with differences in song characteristics while controlling for shared ancestry, we used a published phylogeny containing 9993 avian species[40]. Since nearly all extant passerine species are included in this tree, we were able to utilize all of our mating system, EPP, and song data in phylogenetically controlled analyses. We measured whether closely related species tended to have similar song characteristics using two different methods, Pagel's $\lambda$[41] and Blomberg's $\kappa$[42]; all seven song characteristics showed significant phylogenetic signal with both Pagel's $\lambda$ and Blomberg's $\kappa$ (Supplementary Table 1).

To compare the distribution of song phenotypes between species with different mating systems while accounting for the

statistical non-independence of closely related taxa, we performed a phylogenetic ANOVA (PhylANOVA, R package: phytools[43], based on the algorithm in ref. [44]). We found that no song characteristic was significantly different between monogamous and polygynous species after multiple-hypothesis correction for seven tests with the Holm–Bonferroni method (Fig. 1, Table 3, Supplementary Table 2). We then reconstructed the ancestral states of both mating system and song characteristic and mapped them onto the phylogenies (Fig. 2, Supplementary Figures 1–6). No tests of correlation between mating system and song traits were significant, though we note that our results for syllables per song and song rate trend in the same direction as the results of Read and Weary[30], who found that that polygynous species tended to have a lower song rate and higher mean syllables per song, which they term "syllable repertoire" (Fig. 1).

When testing for correlations between EPP and song characteristics, we find that syllable repertoire is significantly higher in species with low EPP, even when controlling for multiple hypothesis testing (Figs. 1 and 3, Table 3, Supplementary

**Table 3 Results of phylogenetically controlled analyses of mating behaviors and song characteristics**

| | Syllable repertoire | Syllables per song | Song repertoire | Intersong interval | Song duration | Song rate | Song continuity |
|---|---|---|---|---|---|---|---|
| **Mating system** | | | | | | | |
| # of species | 96 | 133 | 145 | 95 | 177 | 91 | 91 |
| PhylANOVA | | | | | | | |
| *p*-Value | 0.25 | 0.0597 | 0.063 | 0.579 | 0.899 | 0.0698 | 0.221 |
| Min; Max | 0.342; 0.236 | 0.028; 0.084 | 0.048; 0.078 | 0.76; 0.496 | 0.952; 0.874 | 0.696; 0.58 | 0.35; 0.1632 |
| Jackknife resampling | None significant | 7/32 families $p < 0.05$ | 5/32 families $p < 0.05$ | None significant | None significant | 2/24 families $p = 0.04$ | None significant |
| Brownie | | | | | | | |
| *p*-Value | 0.0056 | 0.4248 | 0.1478 | 0.4881 | 0.0075 | 0.0266 | 0.3985 |
| Rate higher in | Polygyny | N/A | N/A | N/A | Monogamy | Polygyny | N/A |
| Min; Max | 0.014; 0.008 | 0.042; 0.159 | 0.088; 0.127 | 0.007; 0.646 | 0.024; 0.007 | 0.373; 0.535 | 0.465; 0.384 |
| Jackknife resampling | 25/25 families $p < 0.05$ | 2/32 families $p < 0.05$ | None significant | None significant | 43/45 families: $p < 0.05$ Icteridae: $p = 0.066$ Fringillidae: $p = 0.053$ | 19/24 families $p < 0.05$ | None significant |
| **EPP** | | | | | | | |
| # of species | 57 | 67 | 72 | 45 | 64 | 45 | 45 |
| PhylANOVA | | | | | | | |
| *p*-value | 0.001 | 0.02 | 0.566 | 0.045 | 0.329 | 0.714 | 0.052 |
| Min; Max | 0.004; 0.004 | 0.048; 0.036 | 0.334; 0.656 | 0.098; 0.052 | 0.22; 0.398 | 0.772; 0.518 | 0.064; 0.053 |
| Jackknife resampling | 24/24 families $p \leq 0.015$ | 24/25 families $p < 0.05$ | None significant | 6/17 families $p < 0.05$ | None significant | None significant | 6/17 families $p < 0.05$ |
| Brownie | | | | | | | |
| *p*-Value | 0.1156 | 0.2764 | 0.3792 | 0.3650 | 0.1590 | 0.5785 | 0.3898 |
| Rate higher in | N/A | N/A | N/A | N/A | N/A | N/A | N/A |
| Min; Max | 0.121; 0.166 | 0.477; 0.187 | 0.532; 0.220 | 0.026; 0.408 | 0.142; 0.075 | 0.018; 0.056 | 0.291; 0.421 |
| Jackknife resampling | None significant | None significant | None significant | None significant | 1/25 families $p < 0.05$ | None significant | None significant |

We tested whether song characteristics were significantly different between polygynous and monogamous species and between high and low EPP species (phylANOVA). We also tested whether song characteristics evolved faster in polygynous vs. monogamous lineages or in high vs. low EPP lineages. For each analysis, we assessed the robustness of our findings by testing whether the minimum and maximum values of song characteristics from the literature yielded the same results (Min; Max). In addition, we removed each avian family from the analysis in turn and repeated the analyses (Jackknife resampling), summarized here but reported in full in Supplementary Data 2–3. If the removal of any family led to significant results at the 0.05 level, we note the number of families that met this threshold out of the total number tested

Figures 7–12). These results did not qualitatively differ with a jackknife resampling test, removing each family in turn (Supplementary Data 2). No other song characteristics significantly differed between high and low EPP species after correcting for multiple hypothesis testing; we found a trend toward higher syllables per song in species with low rates of EPP that did not pass the threshold for significance with the Holm–Bonferroni correction (Supplementary Table 2).

**Rates of song evolution under different mating systems.** Next, we assessed whether song characteristics evolved at different rates during polygynous vs. monogamous periods in evolutionary history using the Brownie algorithm[45]. This algorithm tests whether a continuous trait evolves at different rates during the different ancestral states of a discrete trait; for example, whether syllable repertoire size evolved faster in polygynous or monogamous lineages. First, we estimated the ancestral states of a discrete mating system trait—here, polygyny vs. monogamy and low vs. high EPP—and rates of transition across a phylogeny using all oscine and suboscine species for which we had trait data. Using the resulting estimated rates of transition between these states, we generated 1000 unique stochastic character maps to simulate the evolutionary history of these ancestral states on the phylogeny (e.g., Supplementary Figure 13)[43,46]. We then used the Brownie algorithm to test whether mating system affects the evolutionary rate of song characteristics by comparing the average log-likelihood of the one-rate model (i.e., null model: song trait evolution occurs at same rate regardless of mating system) to that of the two-rate model (i.e., the rate of evolution of a song trait differs in monogamous vs. polygynous states: Supplementary Figures 14–15). We did not include runs that failed to converge within 75,000 iterations in our average log-likelihood.

With this procedure, we find that the rate of syllable repertoire size evolution is higher in polygynous lineages than monogamous lineages, a relationship that is robust to jackknife analysis across families and persists when using either the minimum or maximum syllable repertoire values found in the literature (Fig. 4a, Table 3, Supplementary Figure 16, Supplementary Data 3). Thus, polygyny appears to lead to faster evolution of syllable repertoire size, but not to systematically larger repertoires. This rapid but non-directional evolution could indicate that stochastic fluctuations of syllable repertoire occurred frequently in evolutionary history, or that the direction of evolution varies for different ranges of syllable repertoires, such as polygyny driving syllable repertoire toward either extreme or moderate values.

With the same type of analysis, we also observed that song duration evolved significantly faster in monogamous lineages, again without producing an appreciable difference in duration between monogamous and polygynous species (Fig. 4b). This result was also mostly robust to a jackknife analysis: with each family removed in turn, we observed a similar rate distribution, but with the removal of two families, the difference in rates was no longer significant (Brownie Jackknife likelihood-ratio test: Icteridae $p = 0.066$, Fringillidae $p = 0.053$; Supplementary Figure 17, Supplementary Data 3). EPP did not significantly affect the rate of evolution of any song trait, including syllable repertoire (Fig. 4c).

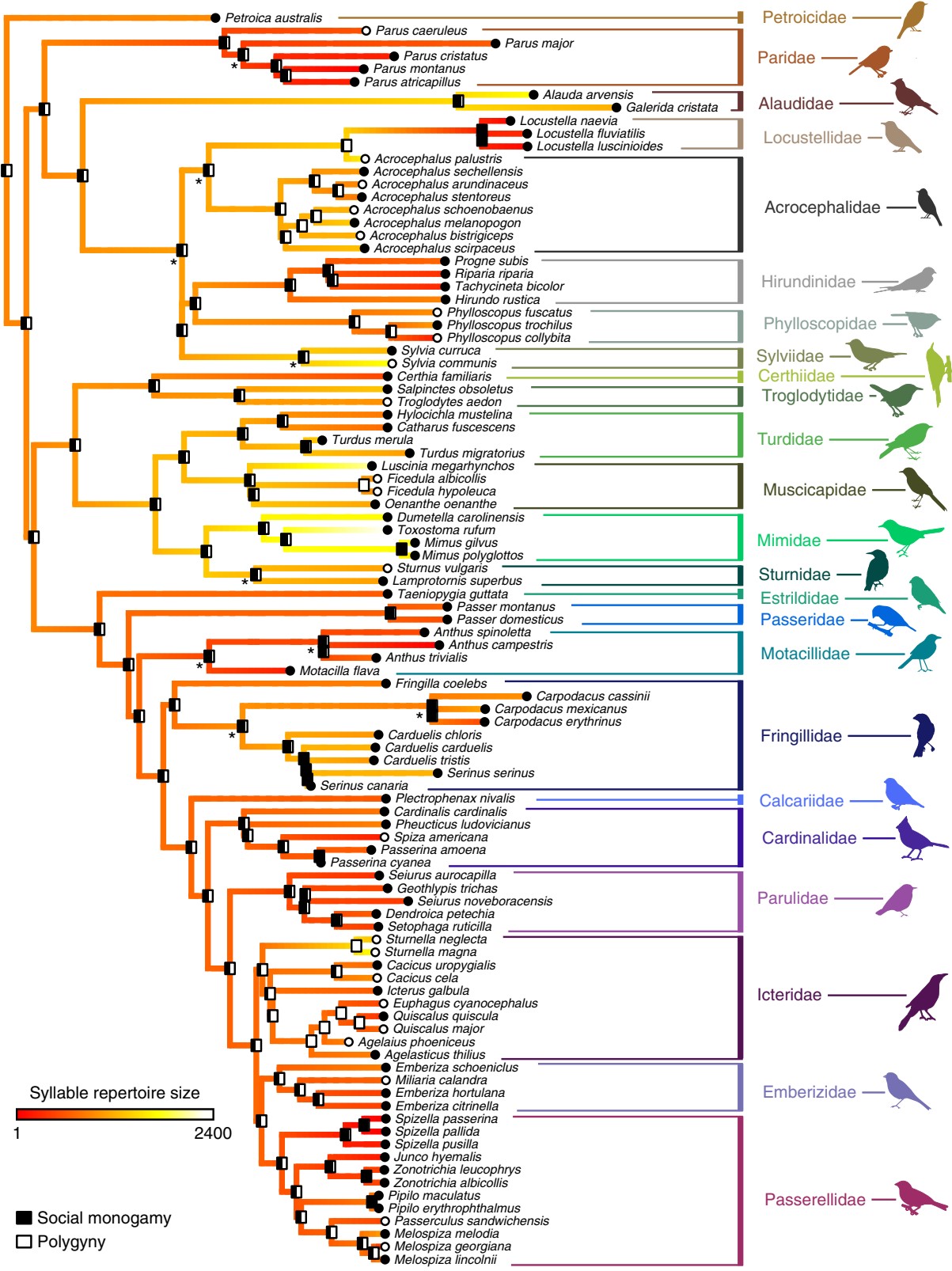

**Correlated evolution between song and mating variables**. Do changes in syllable repertoire size occur independently of mating behaviors, or are song changes more (or less) likely in non-monogamous lineages? To detect non-linear evolutionary trends that might produce the complex patterns of evolutionary rate and direction observed in Figs. 1–4, we devised a method to test for correlated evolution between mating strategy and song

characteristics across the full range of continuous song trait values by separating these continuous values into a series of discrete categories. The results of this analysis to detect correlated evolution (BayesTraits[47,48]) add a layer of complexity that helps explain the first two results. When we break down our syllable repertoire size values into discrete transitions between smaller and larger repertoires, we find evidence for correlated evolution

**Fig. 2** Ancestral character estimation of polygyny and syllable repertoire. At the tips of the tree, monogamy is indicated by black circles and polygyny is indicated by white circles. At the nodes of the tree, bars indicate the results of an ancestral character estimation algorithm, with the black fraction of the bar indicating the percent likelihood that the ancestor at that node was monogamous. The colors along the branches of the tree indicate the estimated ancestral syllable repertoire size. The syllable repertoire sizes ranged from 1 to 2400 in these species and were $\log_{10}$ transformed for analysis. Asterisks indicate nodes that had less than 70% support across 1000 tree replicates; no node had less than 50% support on this tree. Monogamous and polygynous species did not have significantly different syllable repertoire sizes (PhylANOVA $p = 0.250$). Images representing taxa were used or modified from PhyloPic (http://phylopic.org). Several images are used under Creative Commons licenses, with changes made where indicated: Sturnidae (credit to Maxime Dahirel, http://creativecommons.org/licenses/by/3.0/), Estrildidae (credit to Jim Bendon for photography and T. Michael Keesey for vectorization, https://creativecommons.org/licenses/by-sa/3.0/), Fringillidae (credit to Francesco Veronesi (vectorized by T. Michael Keesey), https://creativecommons.org/licenses/by-nc-sa/3.0/), Emberizidae (credit to L. Shyamal, https://creativecommons.org/licenses/by-sa/3.0/; this image was also adapted for Parulidae), and Mimidae and Motacillidae (credit to Michelle Site, https://creativecommons.org/licenses/by-nc/3.0/, both adapted from the original image)

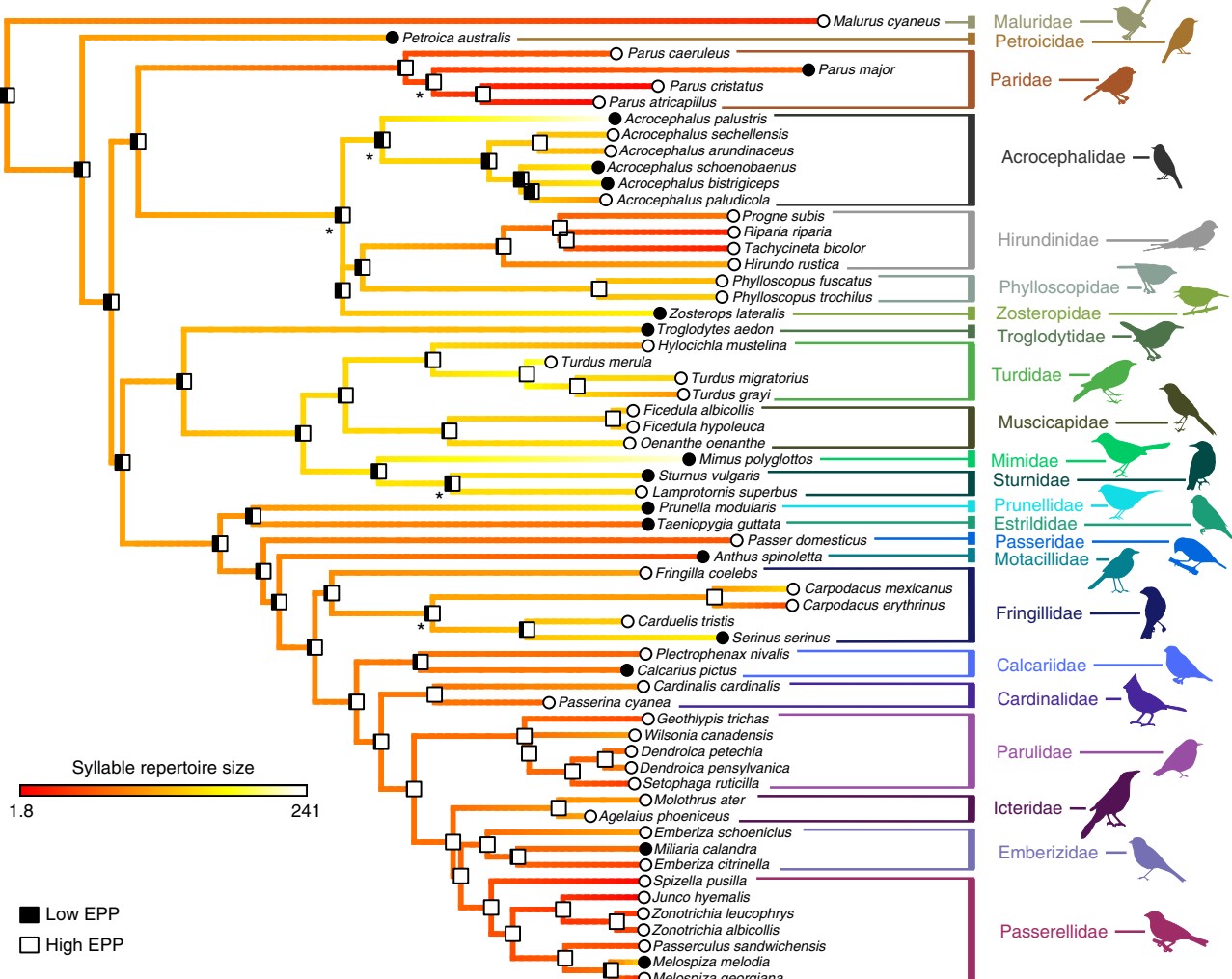

**Fig. 3** Ancestral character estimation of extra-pair paternity and syllable repertoire. At the tips of the tree, low (<10%) EPP is indicated by black circles and high EPP is indicated by white circles. At the nodes of the tree, bars indicate the results of an ancestral character estimation algorithm, with the black fraction of the bar indicating the percent likelihood that the ancestor at that node had low EPP. As in Fig. 2, the colors from red to white along the branches of the tree indicate the estimated ancestral syllable repertoire size. Asterisks indicate nodes that had less than 70% support across 1000 tree replicates; no node had less than 50% support on this tree. The syllable repertoire sizes ranged from 1.8 to 241 in these species and were $\log_{10}$ transformed for analysis. We found that species with high EPP had significantly smaller syllable repertoires than species with low EPP when controlling for phylogeny (PhylANOVA $p = 0.001$). Images representing taxa were used or modified from PhyloPic (http://phylopic.org). Several images are used under Creative Commons licenses, with changes made where indicated: Sturnidae (credit to Maxime Dahirel, http://creativecommons.org/licenses/by/3.0/), Estrildidae (credit to Jim Bendon for photography and T. Michael Keesey for vectorization, https://creativecommons.org/licenses/by-sa/3.0/), Fringillidae (credit to Francesco Veronesi (vectorized by T. Michael Keesey), https://creativecommons.org/licenses/by-nc-sa/3.0/), Emberizidae (credit to L. Shyamal, https://creativecommons.org/licenses/by-sa/3.0/; this image was also adapted for Parulidae), and Mimidae and Motacillidae (credit to Michelle Site, https://creativecommons.org/licenses/by-nc/3.0/, both adapted from the original image)

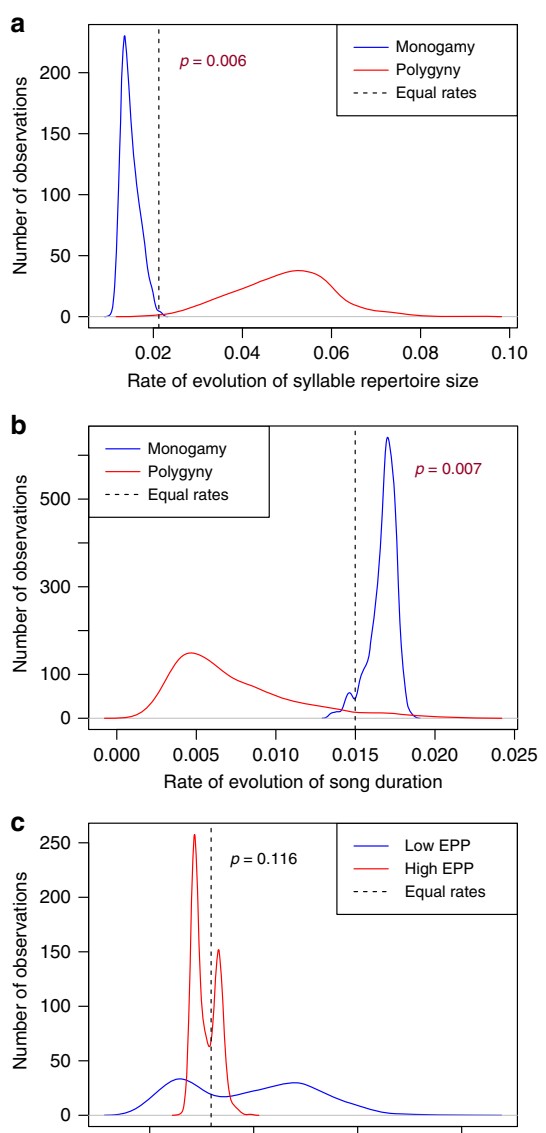

between mating system and syllable repertoire size (Fig. 5): the combination of polygyny and very small syllable repertoires is evolutionarily unstable, with an especially high rate of transition to larger syllable repertoires (Fig. 5a). In addition, polygyny with very large syllable repertoires is also an unstable combination, with species tending toward smaller syllable repertoires (Fig. 5c). Interestingly, we found a qualitatively similar pattern of correlated evolution for each of our studied song characteristics. When combined with polygyny, both very high and very low values of song characteristics were relatively unstable, with elevated rates of evolutionary transitions toward more moderate values of syllables per song, song repertoire size, intersong interval, song duration, song rate, and song continuity (Supplementary Figures 18–23).

We also tested for correlated evolution of song characteristics and EPP. Our results complement our finding that syllable repertoire sizes are larger in species with low rates of EPP: we find patterns of correlated evolution that suggest that the combination of low rates of EPP and small syllable repertoires is unstable, with elevated transition rates leaving this state (Fig. 6). Similarly, there are rapid evolutionary transitions away from the combination of large repertoires and high rates of EPP. These observations agree with the findings that species with low rates of EPP generally have larger syllable repertoires. When we tested for correlated evolution between EPP and other song characteristics, we found a similar pattern only with syllables per song; rates of evolutionary transition were fastest when leaving the combinations of low EPP with low syllables per song and high EPP with high syllables per song (Supplementary Figure 24). For the remaining song characteristics, we observed either no significant evidence for correlated evolution with EPP (song repertoire size, song rate) or qualitatively different patterns from that of EPP and syllable repertoire size (intersong interval, song duration, song continuity) (Supplementary Figures 25–29).

In order to account for variation in tree branch lengths and topology, we repeated PhylANOVA, Brownie, and BayesTraits analyses using a consensus tree computed from trees only containing species with genetic data and using multiple individual trees. These results were consistent with those performed using our original consensus tree (Supplementary Figures 32–35 and Supplementary Table 3).

**Combined effects of mating system and EPP on song evolution.** Finally, we sought to test whether there are interacting effects of mating system and EPP on the evolution of song characteristics. We performed phylANOVAs to test whether there was a significant difference in song characteristics between the different combinations of mating system and EPP: Monogamy + LowEPP ($N = 7$), Monogamy + HighEPP ($N = 31$), Polygyny + LowEPP ($N = 6$), Polygyny + HighEPP ($N = 8$). Syllable repertoire size was significantly associated with species' combined mating system/EPP classification (PhylANOVA $p = 0.0265$). This significant result seems to be attributable to the significantly larger syllable repertoires in the Polygyny + LowEPP group ($N = 6$) compared to the Monogamy + HighEPP group ($N = 31$) (PhylANOVA $p = 0.045$); no other pairwise comparisons were significant (Supplementary Figure 36 and Supplementary Table 4). Next, we used PGLS to test whether song characteristics show a relationship to mating system, EPP, and the interaction between the two. The PGLS recapitulated the result showing High EPP to be correlated with smaller syllable repertoire, but did not show evidence that the interaction between EPP and mating system has influenced syllable repertoire (Supplementary Table 5). Finally, we performed a Generalized Linear Mixed Model (GLMM), which allowed us to both test the interacting effects of mating system and EPP and control for

**Fig. 4** Analysis of the effect of mating system and EPP on the rate of syllable repertoire size evolution. **a** Mating system and syllable repertoire: We generated 1000 stochastic character maps—simulations of the evolutionary history of monogamy and polygyny mapped onto the phylogeny—and then we tested whether syllable repertoire size evolved at different rates in monogamous vs. polygynous branches of the tree. From all runs that converged out of 1000 total runs of the Brownie algorithm, we plot the distribution of the rate of evolution of syllable repertoire size in monogamous lineages (blue) and the rate of evolution of syllable repertoire size in polygynous lineages (red). Distributions are kernel density plots generated using the R function density with a Gaussian smoothing kernel. In all panels, the dashed line indicates the rate of evolution estimated when the song characteristic is assumed to evolve at the same rate regardless of mating behavior. We find that syllable repertoire size evolves significantly faster in polygynous branches (Brownie likelihood-ratio test $p = 0.006$). **b** Mating system and song duration: The rate of evolution of song duration also differed, evolving significantly faster in monogamous lineages. **c** EPP and syllable repertoire: We performed a similar analysis with high and low rates of EPP mapped onto the phylogeny, and tested whether syllable repertoire size evolved at different rates in periods of high (red) vs. low (blue) rates of EPP. We do not reject the null hypothesis that syllable repertoire size evolved at the same rate in high-EPP and low-EPP branches of the tree

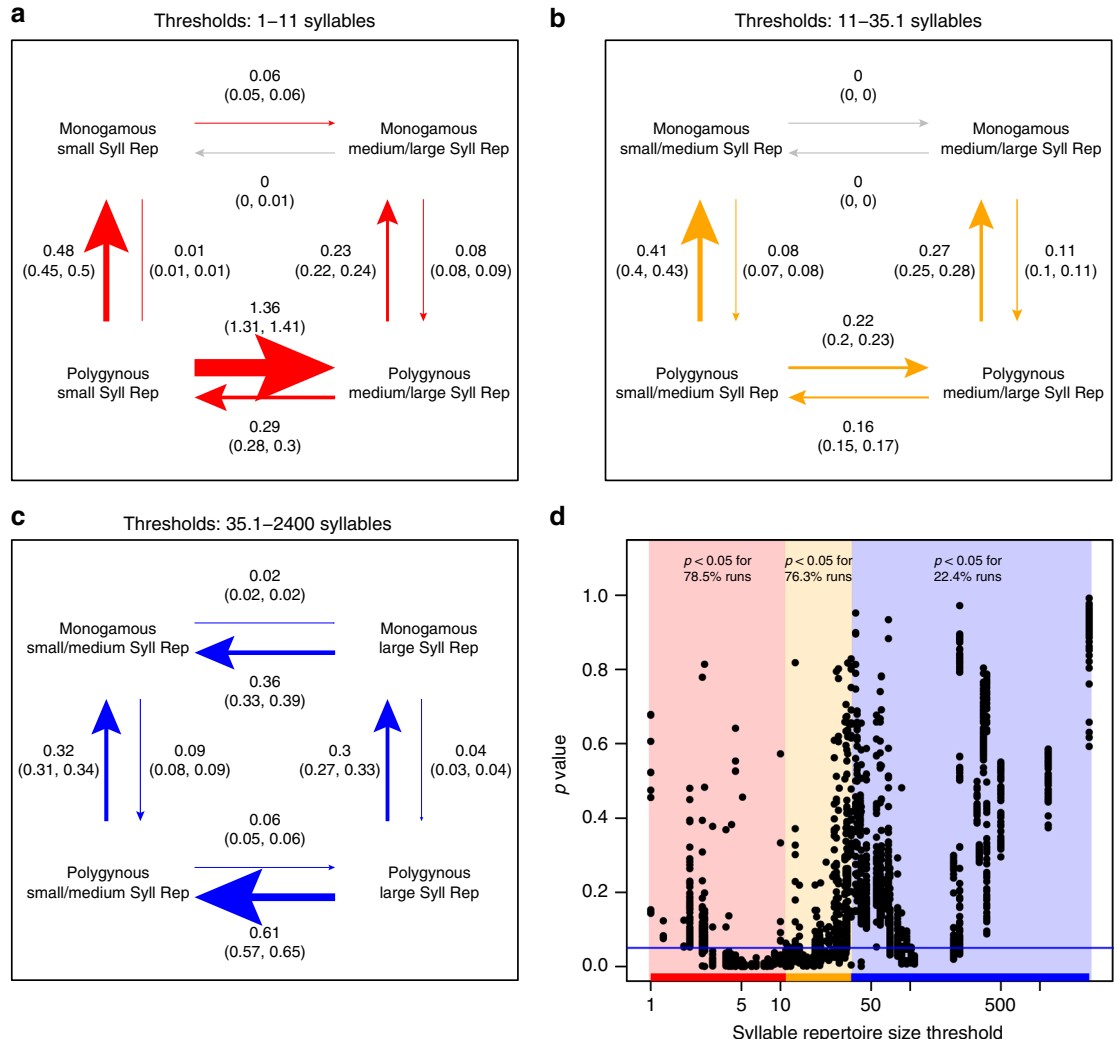

**Fig. 5** Detecting correlated evolution of mating systems and syllable repertoire size. We tested the correlated evolution of syllable repertoire size and polygyny using BayesTraits, with syllable repertoire size made binary based on a threshold delineating smaller vs. larger syllable repertoires. Each value of syllable repertoire was used as the threshold for 100 runs of BayesTraits per threshold. For each plot, there are eight possible transitions between the four trait pairs, shown by arrows. **a–c** We generated transition plots by calculating the mean rate and 95% confidence interval (in parentheses) for each of the eight transitions. **a** When the threshold between smaller and larger syllable repertoires is in the lowest third of observed values (24 unique thresholds ≥1 and <11 syllables, red arrows), polygyny is unstable with small syllable repertoires. **b** When the threshold between smaller and larger syllable repertoires is in the middle third of observed values (24 unique thresholds ≥11 and <35.1 syllables, yellow arrows), the transitions between smaller and larger syllable repertoires do not appear to be elevated in either monogamy or polygyny. **c** When the threshold between smaller and larger syllable repertoires is in the highest third of observed values (24 unique thresholds ≥35.1 and <2400 syllables, blue arrows), the combination of large syllable repertoires and polygyny is unstable, with the highest transition rate pointing to a decrease in repertoire size in the presence of polygyny. These results were robust to jackknife resampling across families (Supplementary Figure 30). **d** For each run of BayesTraits, we performed a likelihood-ratio test to assess whether the model of correlated evolution between mating system and syllable repertoire size was a significantly better fit to the data than the independent evolution model; p-values are plotted against the syllable repertoire size values defining the threshold

multiple song measures from a single species when we had more than one published estimate in our database. This GLMM analysis also reaffirmed our result that EPP significantly predicts syllable repertoire (GLMM, $p < 0.001$), but no other song variable was associated with EPP. Mating system and the interaction between mating system and EPP were not significant predictors of any song variable (Supplementary Figures 37 and 38 and Supplementary Table 5). Finally, the GLMM analysis corroborated the results of our tests for phylogenetic signal: building a linear mixed model with only phylogenetic relationships (without any mating behavior data) led to reasonable predictions of most song characteristics (Supplementary Figure 39).

## Discussion

Here, we assemble a large-scale database of mating system classifications and song characteristics across songbird species, which promises to be a useful tool for future evolutionary studies. By extracting information from published sources, including academic journals and curated field guides, we produce a synthesized database including mating system classification (polygynous vs. monogamous) for 764 species, EPP classification (high vs. low rates) for 142 species, and at least one song characteristic (out of syllable repertoire size, song repertoire size, syllables per song, song duration, song rate, song interval, and song continuity) for 360 species (352 oscine, 8 suboscine) (Supplementary Data 1). We

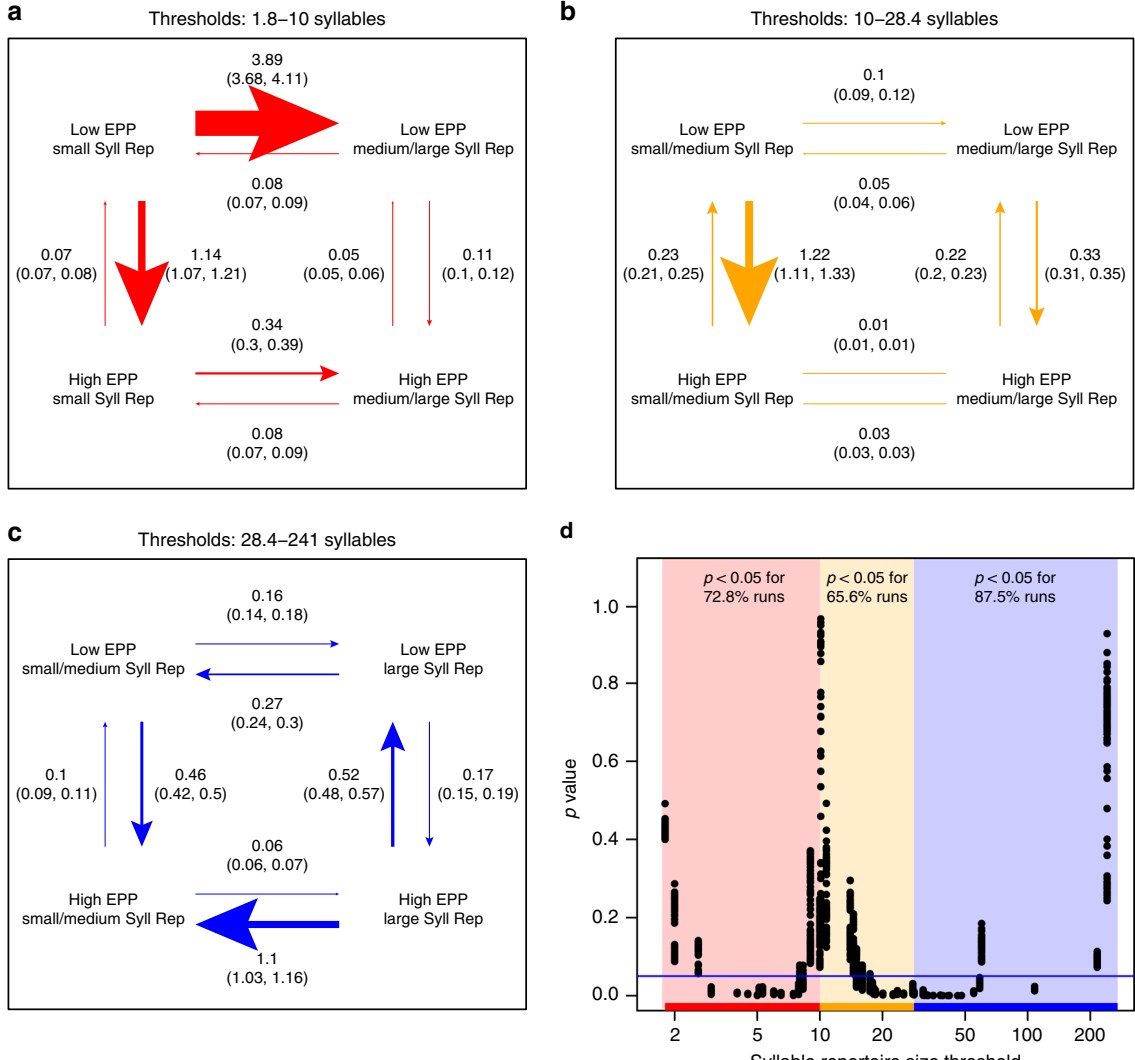

**Fig. 6** Detecting correlated evolution of extra-pair paternity and syllable repertoire size. We tested for correlated evolution of syllable repertoire size and EPP using the same procedure as in Fig. 5. **a** When the threshold between smaller and larger syllable repertoires is in the lowest third of observed values (16 unique thresholds ≥1.8 and <10 syllables, red arrows), low rates of EPP with small syllable repertoires are unstable, and we observe elevated transition rates either toward larger repertoires or toward higher rates of EPP. **b** When the threshold between smaller and larger syllable repertoires is in the middle third of observed values (16 unique thresholds ≥10 and <28.4 syllables, yellow arrows), low rates of EPP are again unstable with small syllable repertoires, and evolutionary transitions toward higher rates of EPP are elevated. **c** When the threshold between smaller and larger syllable repertoires is in the highest third of observed values (16 unique thresholds ≥28.4 and <241 syllables, blue arrows), the combination of large syllable repertoires and high rates of EPP is unstable, and we observe elevated transition rates either toward smaller repertoires or toward lower rates of EPP. We averaged rate values from all runs, regardless of significance. These results were robust to jackknife resampling across families. In the middle segment, only removing Zosteropidae qualitatively altered the dominant rates of transition such that there was accelerated evolution from low to high EPP regardless of syllable repertoire (Supplementary Figure 31). **d** For each run of BayesTraits, we performed a likelihood-ratio test to assess whether the model of correlated evolution between EPP and syllable repertoire size was a significantly better fit to the data than the independent evolution model; $p$-values are plotted against the syllable repertoire size values defining the threshold

synthesize this database with an avian phylogeny and show that song characteristics have strong phylogenetic signal, underscoring the importance of controlling for shared ancestry.

With this database, we perform phylogenetically controlled analyses to assess whether mating strategies alter the evolutionary dynamics of learned song. Since polygyny in a species likely increases the variance in reproductive success and potentially augments sexual selection pressures[4], there is a long-debated hypothesis that polygynous species should evolve more complex or elaborate songs[26,27,29,30,49–51]. We show that, contrary to this prediction, polygynous species overall do not have systematically

larger syllable or song repertoires than monogamous species. However, we find that the rate of evolution of syllable repertoire size is significantly higher in polygynous lineages, but this rapid evolution does not consistently push the syllable repertoire toward increased complexity. Instead, we find that the combinations of polygyny with very small repertoires and polygyny with very large repertoires are both unstable, and polygyny appears to drive song evolution toward moderate values of syllable repertoires (Fig. 5).

When we analyze song evolution in the context of EPP, we find different patterns from what has been previously predicted:

syllable repertoires are significantly larger in species with low rates of EPP, and there is correlated evolution that favors transitions to states combining low rates of EPP with large syllable repertoires and high rates of EPP with small syllable repertoires. These results are in contrast to the recent analysis of 78 species by Hill et al.[39], which presented a phylogenetically controlled regression analysis of EPP percentage and their own measured data from three individual song recordings for each species. With these quite different methods and data types, they reported a positive association between EPP and within-song complexity, but not for song or syllable repertoire size, warranting follow-up studies with more comparable datasets.

Our results generate new questions that necessitate further investigation. First, why do polygynous species appear to evolve moderate-sized song characteristics? One hypothesis could be that the amplified sexual selection of polygyny favors repertoire sizes, and perhaps song characteristics more generally, that strike a balance between being challenging to learn for the male and requiring a low time investment to assess by the female. Second, why might polygyny lead to accelerated rates of evolution of syllable repertoire size and song duration? This result appears to be linked to our observation that very large and very small repertoires, as well as very long and very short songs, evolve to more moderate values in polygynous lineages. The ecological mechanism underlying this evolutionary pattern remains unclear, but, intuitively, we might think of this as the effect of combined forces of two repellant states at either extreme end of the spectrum for a given song characteristic at which selective pressure increases. Third, does EPP increase or decrease sexual selection pressure, and is this effect different in polygynous vs. monogamous species? Our results indicate that, contrary to predictions, EPP seems to decrease the intensity of selection for large syllable repertoire, potentially to the point of selecting against larger syllable repertoires. This result also stands in contrast to our findings regarding mating system-dependent rates of evolution of syllable repertoire. Taken together, this suggests that EPP and polygyny do not have uniform effects on sexual selection of song. Thus, their interactions may produce more complex patterns of selection, leading us to hypothesize that the effects of EPP on variance in male reproductive success is intrinsically dependent upon the dominant social mating system of the species. Alternatively, females in monogamous vs. polygynous species may have different preferences in song or differentially value other features relative to song in social mates vs. extra-pair mates. None of our analyses that combined mating system and EPP yielded significant results for any song characteristic, but this may be due to the relatively low numbers of species that had a combination of mating strategies other than socially monogamous with high EPP. This question should be revisited as mating strategy and song data become available for more species.

It is difficult to directly compare song characteristics across bird species that structure their songs very differently. In order to mitigate these difficulties, we highlight results on syllable repertoire size as a metric that can be interpreted across the songbird lineage. Other metrics, such as syllables per song and song repertoire size, can be influenced both by how a species structures its song and by how human observers define a "unit" of song, making them less ideal for a large cross-species analysis. These metrics may be particularly unreliable for species without stereotyped song bouts, such as Mimid species. Indeed, excluding the Mimidae family from some analyses led to statistically different results (e.g. Mating System + Syllsong: Brownie) compared to when Mimids are included (Supplementary Data 2–3). In addition, for tractability we encoded both mating system (polygyny and monogamy) and rate of EPP (high EPP and low EPP) as binary traits. However, the mating systems of passerines are more

nuanced than is reflected in this study; for example, rates of polygyny can vary widely among polygynous species. Additionally, the rarer mating systems of polyandry, polygynandry, and promiscuity are not reflected in this study.

Further, although song complexity and repertoire size can frequently predict mate choice in laboratory experiments, mate choice in the wild is complicated by numerous other factors. Territory quality, particularly in the sense of food availability, might be particularly salient to potential mates of polygynous males, since the territory might need to support multiple clutches with less contribution to nest provisioning from the polygynous male. Morphological traits such as plumage coloration and sexual size dimorphism are also subject to sexual selection and may play a greater role in mate choice in some species. Song evolution is also made more complicated by other functions of song that may also be selected upon. Some of these functions, such as intrasexual aggression and territory defense, may put similar evolutionary pressures on song as mate choice, but other social functions may not. Finally, the evolution of female song has been historically understudied, but could be correlated with male song evolution while being subject to different selective pressures.

Here, we use phylogenetically controlled computational analyses to demonstrate that mating system and EPP can affect the evolution of song in multiple ways—by influencing the direction of evolution, the rate of evolution, or the likelihood of transitioning to a particular state. In contrast to the long-discussed prediction that polygyny should lead to the evolution of more elaborate songs, these observations suggest more complex, nonlinear dynamics, indicating that the evolutionary associations between non-monogamous mating and sexually selected characters should be analyzed in broader contexts and using methods that incorporate the rate and direction of evolution. In addition, EPP had a significant relationship to song evolution in the opposite direction than has been typically predicted for non-monogamous mating, suggesting that different types of non-monogamous mating behaviors can have dramatically different effects on sexual selection. Building upon these complex dynamics, we propose that EPP and mating system may not influence sexual selection on song independently from one another; further investigation into their interacting effects is needed.

## Methods

**Database assembly**. To assemble the most comprehensive database of mating systems for passerine birds, we compiled information from multiple sources. These included curated field guides, previous studies that investigated mating system evolution, song evolution, or both, and other studies that we found via targeted searches using Google Scholar and Web of Science, as detailed below.

**Mating system and extra-pair paternity data collection**. Birds of North America Online (BNA[52]) is an online encyclopedia curated by the Cornell Lab of Ornithology containing profiles of several hundred avian species that reside in or migrate through North America, including Hawaii. Profiles include life histories compiled from peer-reviewed literature, field guides, and personal observations of profile authors, as well as multimedia from Macaulay Library[53] and eBird[54]. Profiles for 329 Oscine and 34 Suboscine species were available, with varying degrees of documentation. The available search function in BNA only queried species names, so, in order to generate an initial database, we navigated to each species page and used the search page function to find the terms "monogamy"/ "monogamous" and "polygyny"/"polygynous" using the character strings "monogam" and "polygyn", respectively. This search yielded preliminary mating system data for 291 species. These included passages that did not qualify to be included in the final dataset in isolation due to ambiguous wording (i.e., "probably monogamous", "usually monogamous", etc.). If a citation for a digitally available peer-reviewed study was provided, we attempted to find the cited source and other primary literature by searching "(species name)" + "monogam*/polygyn*" (based on each species' BNA entry) in Google Scholar and Web of Science, though we did not remove otherwise definitively classified mating system data for which we could not locate the original source. We searched more broadly for sources that compiled data on species mating systems using Google Scholar and Web of Science with the

search terms "Passeriformes"/"songbird" + "mating system"/"social monogam*"/ "polygyn*"/"extra-pair".

We supplemented data on mating systems and EPP found in traditional publications with species descriptions found in Handbook of Birds of the World Online (HBW[55]). While not peer-reviewed, HBW is rigorously edited on a yearly basis with a demonstrably high standard for data, and several other studies[56–58] have used it as a primary source. We searched for pages using the following search terms with associated yields: (1) polygyn* AND song NOT "presumably polygynous" NOT "probably polygynous" = 135 species; (2) monogam* AND song NOT "presumably monogamous" NOT "probably monogamous" = 826 species. We included "song" as a search term to attempt to limit results to passerine species and manually eliminated search results from non-passerine species. We read the "Breeding" and "Voice" section of each species that was returned in the highlighted searches above to gather any available data on mating system and song characteristics, respectively.

Any species with unambiguous mating system information was included in the final dataset. We also included species described with the terms "primarily monogamous/polygynous". We included species with notes including "apparently" or "appears monogamous/polygynous" only if there was another corroborating source. We used species described as "mostly", "normally", "typically", and "generally monogamous/polygynous" only if quantitative measurements were also given. Species with "Occasional", "Opportunistic", or "Facultative" polygyny were considered monogamous unless other published studies provided disputing evidence.

**Song data collection.** We performed searches in Google Scholar and Web of Science using the queries generated by the following search terms: [species name] + "syllable repertoire", "Passeriformes" + "syllable repertoire", and "songbird" + "syllable repertoire". We then repeated these search queries using "song repertoire" instead of "syllable repertoire". For all searches, [species name] was replaced with the taxonomic and common name of each species for which Birds of North America yielded mating system data. This was supplemented with values obtained via HBW.

**Mating system data classification.** There were often multiple sources for a species' mating system. In the event that sources disagreed, we used the following procedure to decide which mating system to code in our database. (1) If any source gave a percentage of the male population that were polygynous, defined as males with >1 social mates[59], we used that value with a threshold of 5% of sampled males exhibiting polygyny to determine the species mating system; i.e., if <5% of males exhibited polygyny, we classified the mating system as social monogamy; if ≥5% of males exhibited polygyny, we classified the mating system as polygyny. We do not suggest that this threshold value is inherently biologically relevant, but it is consistent with the previous studies[1,23,59–61], allowing us to evaluate our results in the context of earlier research. (2) If multiple sources provided percentages of polygyny in a population that yielded conflicting mating system classification, the median value of those percentages was used. (3) If multiple sources provided qualitative data on mating system that were in disagreement, HBW was used to determine the final mating system classification. If HBW did not mention mating system in its entry for a particular species when there was such a discrepancy, the species was omitted from the dataset.

**Extra-pair paternity data classification.** To determine whether a species exhibited high or low incidence of extra-pair mating, we prioritized sources with genetic parental determination data. A review of EPP studies estimated the cross-species average to be ~11% of offspring per nest to be attributable to extra-pair mates[8]. In line with this estimate and with the previous studies[23], we used a 10% threshold for either extra-pair young (EPY) or nests containing at least one extra-pair chick to estimate the frequency of EPP in that species (<10% = low EPP; ≥10% = high EPP). If a source provided data on both the percentage of offspring in a population that were genetically unrelated to their social father and the percentage of nests that contained at least 1 EPY, we used the former metric, which is more commonly used, to determine species EPP. Either value might be partially determined by cryptic female mate selection, which is a potentially important mechanism by which females may influence which males achieve genetic paternity after extra-pair copulation. Rarely, studies have reported extra-pair mating behavior in terms of observed copulations. We did not include these values for categorization of EPP in our database due to the rarity of the data and since all species for which this was reported also had a reported %EPP. If we found multiple studies with unique %EPP values for a species, we based the classification on the median of all reported %EPP values.

**Controlling for different metrics of extra-pair paternity.** When studies report a rate of EPP, they typically use one of two metrics. The first, referred to as "EPY", "EPF", or just EPP (as used in this manuscript), refers to the percent of the total young in a population that are not genetic offspring of their social father. The second is usually referred to as "extra-pair broods" (EPB), and represents the percent of broods in a population that have at least one egg/chick that is not the genetic offspring of their social father. We prioritized EPP/EPF/EPY values and used EPB values only if the former were unavailable. To test whether the different

metrics of EPP might influence our conclusions, we performed another test using EPP classifications that were inclusive of EPB values. In this method, if a study reported both EPY and EPB for a single population, we used the mean of the values as that study's EPP value, instead of just the EPY/EPF value. We then used the median of all studies, including studies that only reported EPB, to obtain the EPP value we used in the 10% threshold classification of the species. Using this method did not change the result of phylANOVA for EPP and syllable repertoire, the only test involving EPP that yielded a significant result, nor of any other test.

**Song data classification.** Nomenclature for song characteristics is variable across sources. For example, in some sources, "syllable repertoire" means the average total number of syllables an individual uses in song; however, several others, including Read and Weary[30], have considered this term to mean "average number of unique syllables per song". We checked the primary source methods whenever possible to ensure we classified each song parameter correctly. We categorized several measures that have generally been used as proxies of song complexity, elaborateness, or variety, as follows: (1) Syllable repertoire size (Syll Rep), defined as the average total number of unique syllables an individual uses across songs. Some studies also called this "song repertoire", so we read the relevant methods sections for clarification. (2) Syllables per song (Syll/song), the average number of unique syllables used per song (also occasionally called syllable repertoire, e.g., ref.[30]). (3) Song repertoire size (Song Rep), the average total number of unique songs an individual produces. We also categorized song metrics that have traditionally been used as proxies of song performance, as follows: (4) Intersong Interval (Interval), the average length of time separating songs within a period of consistent singing behavior, measured in unit: seconds. (5) Song duration (Duration), the average length of a song, defined as the number of seconds of consistent singing between periods of silence; sources may have differed in this definition. (6) Song rate (Rate), the number of full song cycles produced per minute, calculated as $60/(\text{Duration} + \text{Interval})$. (7) Song continuity, the proportion of total song performance time spent producing song, calculated as $\text{Duration}/(\text{Duration} + \text{Interval})$.

While we included other song characteristics when available, we only actively searched for species' syllable repertoires and song repertoires. Several sources and previous studies provided these other metrics in addition to repertoire or mating system data. Any value we encountered during this search we recorded in Supplementary Data 1, but the database should not be considered exhaustive. For values of syllable repertoire obtained from Moore et al.[62], we used only the value from the single study they utilized, since they were systematic in their selection of data. Occasionally we could infer the syllable repertoire of a species if we had data for "syllables per song" and the song repertoire was equal to 1.

If any song data was given for any species as a range, we used the median value. If multiple sources were found for a particular species, we used the median value across sources and noted in the dataset the minimum and maximum values observed for that species. We log$_{10}$-normalized the data for all song characteristics. Some species, in particular those belonging to the Mimid family, do not have discrete, stereotyped songs, instead improvising their songs for an extended, often uninterrupted period of time. Occasionally these species had measured values for syllable repertoire, syllables per song, and/or song repertoire in the literature, but more often they were qualitatively described as "large". Previous studies on song evolution included these species in quantitative analyses by assigning an arbitrarily high value to these characteristics ranging from 100 to 1000, depending on the study[23,30,38,63]. For species that were assigned "large" or arbitrary repertoires in the literature and had no other quantitative assessment of repertoire available, we assigned syllable repertoire, song repertoire, syllables per song, and song duration to have a minimum value of 100, a maximum value of 1000, and a median value of 500.

**Assembling phylogenies.** We assembled two phylogenies for use in our analyses: one including only avian species which had genetic sequence data integrated into the tree and one which also integrated species without genetic sequence data. For the latter, we obtained 1000 trees, each containing 9993 species from Birdtree. org[40,64,65] by randomly choosing one of ten sets of 1000 trees available in the BirdTree data downloads. These trees (HackettStage2 7001–8000) were built by generating relaxed-clock molecular trees for each of 158 clades, then arranging these clades on an avian-wide tree using the backbone determined in Hackett et al.[66] (full methods in Jetz et al.[40]). We used consense in Phylip[67] to generate a consensus tree that specified how many out of the 1000 input trees supported each node (full consensus tree in Supplementary Data 4–5). We used the function consensus.edges (R package: phytools, method: mean.edge) to compute a consensus tree with branch lengths from the 1000-tree sample. For any nodes that resulted in multifurcation, we used function multi2di (R package: ape) to randomly reassign multifurcations as a series of dichotomous nodes, resulting in a bifurcating tree necessary for trait analyses. The edges generated through this method are defined by default as having a length of zero, which the statistical tests below cannot process. We set these edge lengths to an arbitrarily very low value of $10^{-19}$. (We found no qualitative differences in our results with different values of this branch length; see Supplementary Information). For more efficient computation, we used drop.tip (package: ape) to eliminate species for which we had neither song characteristics nor mating system data. For subsequent analyses comparing mating system and song characteristics, we dropped tips of species with no data for a given

comparison (see code in Supplementary Software)[66]. We repeated this procedure using a set of 1000 trees with only the 6670 species for which Jetz et al.[40] had a genetic sequence (HackettStage1 7001–8000). To ensure that results were robust to differences in branch lengths and tree topography, we repeated all analyses using this Gene Tree, as well as on each of the first 100 trees in HackettStage2 7001–8000 for PhylANOVA and Brownie, and the first 20 trees for BayesTraits.

**Phylogenetic comparative analyses**. We used a set of phylogenetic comparative analyses to determine the relationship between the evolution of mating systems and each song characteristic in turn. First, we performed an ancestral character estimation[68] using the ace function (R package: ape) for each trait separately. We calculated the ancestral state at each node for each trait, both continuous (song variables) and discrete (mating system classification, EPP). We used this to visualize the likelihood that each trait was present at each ancestral node of the tree.

To assess the phylogenetic signal of each song characteristic, we calculated both Pagel's $\lambda$[41] and Blomberg's $\kappa$[42] using the function phylosig in phytools[43]. For Pagel's $\lambda$, we assessed significance with the included permutation test, resampling 100,000 times. For each song characteristic, we analyzed all of the species for which we had data for that characteristic, even if we did not have corresponding mating system data for that species.

To assess whether bird species had significantly different song characteristics dependent on their mating strategies, we used the function phylANOVA (R package: phytools[43]) to determine whether song traits were significantly different between polygynous and monogamous species or between species with low EPP and high EPP, controlling for phylogeny. We also calculated the residuals of the ANOVA (Supplementary Figures 40–41).

Next, we used the Brownie algorithm[45] to assess whether the rate of evolution of song characteristics was significantly different between polygynous and monogamous branches of the phylogeny. First, we map the evolution of mating system onto the phylogeny and test whether evolutionary transitions between monogamy and polygyny occur at equal rates (Equal Rates (ER) model) or whether the transition from monogamy to polygyny occurs at a significantly different rate than the transition from polygyny to monogamy (All Rates Different (ARD) model). To do this, we calculated the rates of evolution of mating system across the full tree and dataset using ace (R package ape) using both an ER and ARD model to estimate transition rates. We performed an ANOVA (R package: stats) on the two models to determine whether the ARD transition rate model was a significantly better fit. We similarly tested the transitions between high and low states of EPP. The ARD model was a significantly better fit for polygyny (ANOVA $p = 6.05 \times 10^{-19}$; log-likelihood ER $= -302.59$, ARD $= -263.06$), and EPP (ANOVA $p = 0.0140$; log-likelihood ER $= -84.49$, ARD $= -81.47$) so we recorded the transition rates from the ARD model (monogamous to polygynous: 0.0962, polygynous to monogamous: 0.0114; low EPP to high EPP: 0.0407, high EPP to low EPP: 0.0193) and used them to generate the stochastic character maps (simmaps)[46] for all subsequent Brownie tests. For each song trait, we generated 1500 simmaps from the subsetted tree and dataset for that mating system/song trait combination using the rates derived from the full tree.

We then ran brownie.lite (R package: phytools) once per simmap. For each run, this function calculates the log-likelihood of an ER model (i.e., the song trait evolves at equal rates regardless of the mating system state the ancestral species occupies) and ARD model (i.e., the song trait evolves at a rate dependent on an ancestral species' mating system). To determine whether the ARD model is a significantly better fit than the ER model, we calculated the average log-likelihood of the one-rate model and the average log-likelihood of the two-rate model over our 1500 runs and performed a likelihood ratio test, assessing significance with the function pchisq in R. For some runs, the Brownian motion run does not converge by the time it reaches the maximum number of iterations (we used maxit = 75,000, increased from a default of 2000). If the model did not converge or did not finish in the maximum runtime (16 s), we discard these runs from our final analysis and do not include them in the average log-likelihood.

Currently, there is no published method to test for correlated evolution between a discrete trait and a continuous trait, to our knowledge. We developed a method to utilize the maximum likelihood test of dependent trait evolution of two discrete traits available in BayesTraits to test for correlated evolution between mating strategy and song across the full range of continuous song trait values. This algorithm calculates whether the rate of evolution of one trait is dependent on the state of another trait. For example, we can use this algorithm to ask whether an evolutionary change in mating system alters the likelihood of an evolutionary change in a song characteristic. BayesTraits compares current species data with simple and complex continuous Markov models of evolution of discrete traits on a given phylogeny to determine whether the complex model (i.e., dependent trait evolution) describes the data sufficiently better than the simple model (i.e., independent trait evolution) to justify accepting the complex model. BayesTraits reports marginal likelihoods for the complex and simple models (function Discrete in package btw). We used function LRtest (package: lmtest), which returns the likelihood ratio statistic (LRstat) and p-value, to perform the likelihood ratio test to determine whether to accept the complex model of dependent evolutionary rates over the simpler model of independent evolutionary rates. Since this model requires both traits to be binary, we classified continuous song characteristics as binary groups (low or high) based on a delineating threshold. Instead of choosing

the threshold arbitrarily, we used R package btw (BayesTraits Wrapper) to perform the tests using each unique value of the song characteristic data as the threshold, repeating the test 100 times at each threshold. We plotted the p-values over the threshold used for all tests (Figs. 5d and 6d).

Each repetition of the test analyzed whether the evolution of mating system and song were correlated by performing a log-likelihood comparison between a continuous-time Markov model where the traits evolve independently and a model of correlated evolution, in which the evolution of one trait is dependent on the state of the other trait. BayesTraits Wrapper function btw returns the computed rates of transition between the four total states and a p-value indicating whether we can reject the null hypothesis that evolution of the two traits occurred independently. The computed transition rates vary dramatically across the range of song trait values, depending on the threshold delineating the low and high value categories. When the threshold is set as a lower value, the test effectively evaluates the rate of song values switching from very low to moderate and vice versa. When it is set as a higher value, the calculated rates are the rates of switching between very high song trait values and medium-low values. To account for this nuance, we segmented the vector of unique song trait values into bins and calculated the mean and 95% confidence interval of each state transition rate from all of the runs across all of the thresholds in each bin. We present the results with the thresholds binned into three groups in the main text, but also include the analyses with the thresholds divided into two, four, and five groups in Supplementary Figures 42–47.

**Accounting for variation in measurements**. To account for variation in song characteristic measurements across studies, we repeated PhylANOVA and Brownie calculations with the minimum and maximum values of each species' song characteristics obtained from the literature. For these runs, any species for which we had only one data source for the song trait in question kept that trait value. We replaced the median values of any species with multiple sources with the minimum and maximum trait values, respectively, for a total of three runs per song characteristic. We calculated the values for song rate and continuity in our database based on the median song duration and intersong interval values in the literature. For both song rate and continuity, the minimum and maximum values were calculated using the maximum duration and interval values and minimum duration and interval values, respectively.

**Testing for an interaction between mating system and EPP**. We performed a phylogenetic generalized least squares (PGLS) analysis using the function gls (R package: nlme[69]), computing phylogenetic correlation using the function corBrownian (R package ape). This technique enables us to analyze of whether there is a relationship between song characteristics and multiple variables at once: mating system, EPP, and the interaction between mating system and EPP.

We also performed a GLMM for each song characteristic using function MCMCglmm (R package: MCMCglmm[70]), which allowed us to include all values found in the literature for each species. For this GLMM, we subsetted the data to include only those species for which we had both mating system and EPP classifications, which also allowed us to evaluate the interacting effects of mating system and EPP. We verified the computed model using posterior predictive checks using functions predict and simulate in the package MCMCglmm.

**Controlling for differences between families**. To ensure that our Passeriformes-wide findings were robust and not driven by any particular family, we performed a jackknife analysis in which we reran all analyses on a series of subsetted datasets with each family removed in turn. We determined the family of each species based on its classification in the 2017 version of the eBird Clements Integrated Checklist[71].

**Testing the effect of multifurcations on our results**. The avian phylogeny constructed by Jetz et al. contains numerous multifurcations, which we observed both in the sample of 1000 trees that we extracted from their tree distribution and in the consensus tree we constructed from those 1000 trees. Since a dichotomous, bifurcating tree is required for phylogenetic comparative analyses, we used the function multi2di (package: ape) to restructure multifurcated nodes on the consensus tree into randomly organized dichotomous nodes with arbitrarily small branch lengths between them. Since the bifurcations are assigned randomly, we repeated the multi2di procedure 10 times with an arbitrary branch length of $10^{-19}$ and 10 times with an arbitrary branch length of $10^{-4}$. Then, we repeated our phylANOVA and Brownie analyses with each of these trees. None of these computed differences led to qualitative differences in the results of any test.

**Reporting summary**. Further information on experimental design is available in the Nature Research Reporting Summary linked to this article.

**Code availability**. The authors declare that all code used in the analysis of data within this paper are available in the associated Supplementary Information files (Supplementary Software). Code is also available at GitHub.com/CreanzaLab/MatingBehaviorsAndSongEvolution.

## Data availability

The authors declare that all data supporting the findings of this study are available within the paper and its Supplementary Information files. Data are also stored on GitHub.com/CreanzaLab/MatingBehaviorsAndSongEvolution.

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

## Acknowledgements

The authors thank Abigail Searfoss, Cristina Robinson, Parker Rundstrom, Jacob Steenwyk, Jennifer Rand, and members of the Creanza lab for valuable feedback.

## Author contributions

K.T.S. and N.C. designed and developed the experiments. K.T.S. compiled the data and wrote the analysis code. K.T.S. and N.C. analyzed results, made figures, and wrote the paper.

## Additional information

**Competing interests:** The authors declare no competing interests.

