## [Peer Review File · Nature Communications]

Reviewers' Comments:

Reviewer #1:

Remarks to the Author:

This study nicely illustrates the complexity of the effects of social selection on signal elaboration. As the authors correctly state, ornithologists have long hypothesized that sexual selection should generally act to increase song complexity—a hypothesis that is invoked often, yet has had mixed support when tested explicitly. This study goes some ways towards resolving these issues by showing that evolution does seem to proceed more rapidly in polygynous systems, but it does not proceed consistently towards more elaboration. The study is unusually thorough in its scope for a study of song evolution, as it looks at multiple measures of song complexity/elaboration as well as multiple measures of mating system (monogamy/polygyny and high/low EPP). The authors appear to have made sincere efforts at validating the robustness of their conclusions by using different methods to check the effects of different phylogenetic structure, sampling, and data classification.

This is not to say that the study is flawless—as with any comparative study, there are limitations to the data collection and analyses. I try to point out where the authors could address some lingering concerns. Moreover, the writing tends to err on the side of over-explanation (which is certainly easier to fix than the opposite problem). I think there are many places where the writing could be made more economical, which might make the manuscript more impactful in the end. I try to point some of these places out in minor comments, but the authors could do more throughout.

General Comments:

I am left wondering about the contrasting results for monogamy/polygyny and EPP. The central hypothesis the study is addressing is about the effects of sexual selection on song elaboration, and these are two mechanisms that might increase variance in reproductive success of males (thus increasing opportunity for sexual selection). Polygyny and EPP are used as two complimentary indices for the opportunity for sexual selection, and thus the readers (at least this reader) is left scratching their heads when the results indicate very different relationships between these two factors and song evolution. However, there was little discussion about how we should reconcile these differences in results. I suspect there is opportunity in the discussion section for a deeper dive into the potential differences between polygyny and EPP as mechanisms for increasing reproductive success in males, as well as the potential differences in covariation between song traits and male reproductive success in these two contexts. On a related note, it also seems possible that the impact of EPP differs between socially monogamous vs. polygynous systems, hinting at interactions between these factors in song elaboration. Ideally, there would be a way to combine both EPP and mating system information to generate an estimate of the opportunity for sexual selection that can be mapped onto the phylogeny, though I would think this might be quite a difficult dataset to assemble. (I note that some of these issues are discussed lightly at the end, in Lines 539-549, but perhaps it could be highlighted more).

Abstract:

The abstract gives short shrift to the results of the analyses with respect to EPP. If the central goal of the study is to test the effects of sexual selection (as measured by reproductive variance in males) on song evolution, then EPP should at least be mentioned.

Introduction:

The two concepts of polygyny and EPP are introduced separately (in the first and fourth paragraph,

respectively), sandwiching a discussion of song complexity. This is in part due to the structuring of the first paragraph around Darwin. While this works for emphasizing the importance of mating systems (polygyny in particular) in evolutionary theory, it does not provide room for discussing EPP (because Darwin didn't know about it). Thus, I wonder if this structure does the study disservice by not providing the logical connection between testing both polygyny and EPP in this study.

Line 55-57. This sentence starting with "Song in oscine songbirds could be a particularly salient target for sexual selection...": This statement is correct and interesting, but seemed out of place as the end to the paragraph specifically about syllable repertoire size.

Lines 185-203: While I understand the reasoning for including these nonparametric tests (i.e., to illustrate what the results might look like using phylogenetically uninformed methods), I think it perhaps takes the reader on a detour that detracts from the main findings based on a phylogenetic methods. One compromise I would propose is to utilize the fact that Figure 1 already visually illustrates what the results look like without phylogenetic controls. So perhaps the authors could simply add a sentence in the previous paragraph (presenting results of the first row in Figure 1) pointing out that this figure shows how, even when some metrics of song complexity appear to have higher values in polygynous mating systems, phylogenetically controlled tests show that mating system has no effect. This would then flow straight to the results with respect to EPP without detouring to the "what the results would've looked like if we had used poor stats".

Lines 164-166: This caveat about the Jetz tree can probably be removed without sacrificing clarity. I'm not sure that anyone will fault the authors for not using a global phylogeny of birds that are not yet based on full genomes at this point.

Lines 173-175: I think this sentence could be removed without sacrificing clarity—I think the point that controlling for phylogeny is important is sufficiently clear already.

Line 268-274. These two sentences flip back and forth from saying the null hypothesis was tested, then saying the alternative was assessed. Perhaps it could be shortened to one sentence, e.g.: "We then used the Brownie algorithm to test whether mating system affects evolutionary rates by comparing the average log-likelihood of the one-rate model (i.e., null model: evolution at same rate regardless of mating system) to that of the two-rate model (i.e., rate of evolution differs in monogamous vs. polygynous states: Supplementary Figs. 14–15)."

Lines 448-451: While it is always true that there are multiple potential evolutionary dynamics. However, it would just be simpler to state the evolutionary dynamic that you detect. Thus, you could delete this sentence and start the paragraph with the next sentence.

Lines 483-505: This paragraph is filled with speculations, which is not very strong. I wonder if it would be space better occupied by a paragraph reconciling the difference between polygyny and EPP, as mentioned in one of my first comments.

Lines 507-523: This paragraph is somewhat repetitive with the paragraph following it, as they both deal with limitations of the approach for interpretation. I think you could delete this paragraph, and then incorporate a couple of points in the following paragraph (e.g., one sentence about how including/excluding Mimidae seems to have an outsized effect).

Lines 529-531: This sentence is repetitive from paragraph above it.

Figures 5 & 6: I think I understand this figure, but I am a bit confused about the arrow sizes. The

width of the line and the size of the arrowhead seems decoupled, with some arrows having a larger head than arrows associated with larger values (e.g., Figure 5b, arrow from poly/medium to poly/small has larger head than arrow from poly/small to poly/medium, despite having smaller transition value). Does this signify something, or is it just a coding mistake?

Reviewer #2:

Remarks to the Author:

This is a well-written, thorough investigation into two main hypotheses thought to drive the evolution of elaborate birdsong - a very complex trait nearly ubiquitous throughout songbirds. The authors create and use a large, comprehensive data set to evaluate their predictions, in one of the most direct assessments of this question that I have seen for some time. Their results are counter to expectations that song is directionally selected in systems in which sexual selection is expected to be high.

The methods are robust. I have a number of logistical issues:

1. The authors should de-emphasize their focus on the non-phylogenetically controlled analyses - as this is inappropriate practice.
2. At each section in which the song metrics are discussed, it is never clear to me where these data come from and it is not clear until the methods how they are classified. Some mention of this sooner and more detail in explaining compilation of these metrics is important, since this is the focus of the paper.
3. The authors taut a large data set, but because they only have song data for ~350 species, this limits the analysis to this smaller sample size (which is still good for this hard-to-come-by song data).
4. There is very little discussion of alternative functions of song or trade-offs with other selection pressures that may be at play. It would be nice to have a paragraph in the discussion dedicated to explanations surrounding these obvious reasons for why the predicted trend was not found (especially given high phylogenetic signal).
5. Discussion, overall could be shorter - getting right to explanation for why the authors believe they see these results.

Specific comments follow:

Line 19: the authors state here that their analysis includes the full songbird phylogeny, however, their data on mating system, song, and EPP only exists for a portion of the songbirds.

Line 20: syllable repertoire size classified how?

End of abstract: Why do you think these two contrary results occur? What implications does this have for how we think about and study bird song?

Line 46: "courtship song" assumes that song in songbirds is used exclusively for attracting mates and/or that you focused exclusively on courtship song.

However, this would be difficult, as song likely serves multiple functions in a lot of species (especially non-polygynous, year-round territorial species) and you may not know the function a priori to sampling. In fact, the multi-functionality of song could be a reason for your unexpected results. Try to avoid specifying concretely the function of song prior to analysis. Also, discussing some of these alternative functions (e.g., territory defense and mediating social interactions) could be worth discussing in intro to refer to later.

Line 121: use the term "shared ancestral history" throughout. Can state "shared ancestry" or "phylogeny" instead.

Line 149: please explain a bit about where these songs came from, any structure to the data (multiple individuals, geographic locations, subspecies per species, etc). How did you select the species you included?

Lines 149-150: please explain how these song metrics (syllable repertoire size, song repertoire size, song interval (what is this?), and song duration) were calculated/quantified. If these values came from

the literature, how was it ensured that these metrics had approximately similar meaning/were assessed similarly among species? Differences in authors' interpretation or calculation of repertoire sizes could lead to different estimates.

Line 152-153: How did you arrive at this sample size? Above you state you have song data for 352 oscines. Below you state you have data for 39 suboscines. How does this total 899? The function of this data set seems limited by the species with song data, as you need the song data to conduct the comparative analyses. So, even if you have more species with mating system or EPP data, the total number of species in the analysis should be dictated by those with song metrics. 300+ species with repertoire data is still impressive.

Please state how many species were included in the comparison to mating system and EPP, respectively.

Line 165: Only species with genetic data in the Jetz tree should be used for certain kinds of comparative analyses (especially those involving rate), which you can specify when downloading trees. Did you take this into consideration?

Line 170-171: this is an odd way to state this. The controlling for phylogeny controls for the statistical non-independence that potentially exists because some species are more closely related than others. Any more, this is standard practice and does not require much explanation.

Line 174-175: a number of studies have shown that it is important to control for phylogeny whether phylogenetic signal is high or not (because of the need to control for the statistical non-independence; and that these metrics represent the signal in the residuals and not necessarily the traits themselves). To avoid promoting improper practice, I would remove this sentence.

Which traits are these calculations for ?

Line 180: How many anovas did you perform? One for each song feature? Did you correct for multiple comparisons?

Why didn't you use ppls or phylogenetic mixed models or other analyses that would have allowed you to simultaneously test for multiple predictors that might influence mating system or EPP concurrently? Such models (esp phylo mixed models) would also allow you control for repeated measures (inclusion of multiple samples per species or individual). Did you have any data structure like this? Again, more explain about this dataset, where it came from, and its structure are needed.

Analysis: did you have any repeated measures for individuals or species? Did you control for this in your model structure?

Lines 185-191: this is not standard practice (analyses should not be run to see what results would be if you did not control for phylogeny, as the analysis controlling for phylogeny is the correct statistical test). You can include this in the discussion, but I would not include it here.

Lines 198-200: again, too much focus on controlling vs not controlling for phylogeny

Line 213-214: You do not discuss differences between controlling for phylogeny here (which is fine, as this should not be a major focus). If you discuss this in the above paragraph, put this statement there.

Figure 1: this is the first you mention how you scored polygyny. Include in text.

Because you should not run phylogenetically uncontrolled analyses in your situation, I would not present the Wilcoxon non-parametric results here. Alternatively, you could highlight marginally significant values, if you think it appropriate.

What is continuity? Mention in text.

Lines 264-266: did you run these analyses on multiple alternative versions of the tree ?

Line 427: "by extracting information from published sources"... This is the first time you mention this. Mention sooner.

Lines 448-481: a lot of repetition here - re-explaining the findings, which were just explained in previous sections. Given short format of this journal, consider cutting or greatly reducing text here to quickly re-emphasize the main findings.

Line 483: this is what the reader is waiting for. get to this important paragraph faster.

Methods could be organized a bit more clearly to point to sections relevant to assembly of each of the

three data sets

Line 674: to me, this appears to be the start of the section explaining song variables, but it is still not clear up front where these data came from...

We sincerely thank the reviewers for their detailed assessment of our manuscript. We think that the current version has been greatly improved by their feedback. We present responses to specific comments below.

Reviewer #1 (Remarks to the Author):

This study nicely illustrates the complexity of the effects of social selection on signal elaboration. As the authors correctly state, ornithologists have long hypothesized that sexual selection should generally act to increase song complexity—a hypothesis that is invoked often, yet has had mixed support when tested explicitly. This study goes some ways towards resolving these issues by showing that evolution does seem to proceed more rapidly in polygynous systems, but it does not proceed consistently towards more elaboration. The study is unusually thorough in its scope for a study of song evolution, as it looks at multiple measures of song complexity/elaboration as well as multiple measures of mating system (monogamy/polygyny and high/low EPP). The authors appear to have made sincere efforts at validating the robustness of their conclusions by using different methods to check the effects of different phylogenetic structure, sampling, and data classification.

This is not to say that the study is flawless—as with any comparative study, there are limitations to the data collection and analyses. I try to point out where the authors could address some lingering concerns. Moreover, the writing tends to err on the side of over-explanation (which is certainly easier to fix than the opposite problem). I think there are many places where the writing could be made more economical, which might make the manuscript more impactful in the end. I try to point some of these places out in minor comments, but the authors could do more throughout.

Thank you for your helpful feedback! We have revised the text to be more succinct throughout. We have also removed certain passages that were repetitive and expanded upon the ideas in the areas suggested.

General Comments:

I am left wondering about the contrasting results for monogamy/polygyny and EPP. The central hypothesis the study is addressing is about the effects of sexual selection on song elaboration, and these are two mechanisms that might increase variance in reproductive success of males (thus increasing opportunity for sexual selection). Polygyny and EPP are used as two complimentary indices for the opportunity for sexual selection, and thus the readers (at least this reader) is left scratching their heads when the results indicate very different relationships between these two factors and song evolution. However, there was little discussion about how we should reconcile these differences in results. I suspect there is opportunity in the discussion section for a deeper dive into the potential differences between polygyny and EPP as mechanisms for increasing reproductive success in males, as well as the potential differences in covariation between song traits and male reproductive success in these two contexts. On a related note, it also seems possible that the impact of EPP differs between socially monogamous vs. polygynous systems, hinting at interactions between these factors in song elaboration. Ideally, there would be a way to combine both EPP and mating system information to generate an estimate of the opportunity for sexual selection that can be mapped onto the phylogeny, though I would think this might be quite a difficult dataset to assemble. (I note that some of these issues are discussed lightly at the end, in Lines 539-549, but perhaps it could be highlighted more).

Thank you for this important perspective. We have made several changes to attempt to more clearly compare and contrast mating system versus extra-pair paternity and to increase the space dedicated to EPP. In the Introduction, we placed the section discussing EPP directly after the section discussing polygyny, as opposed to several paragraphs later. In addition, we investigated further the existing empirical literature regarding the variance in male reproductive success in species with high extra-pair paternity and interactions with polygyny and reference three studies that support different predictions. (Byers et al 2004, Lebigre et al 2012, Vedder et al 2010)

- We agree that the interaction between mating system and EPP is a fascinating and worthwhile question. With the caveat that the sample sizes are small for combinations of mating traits other than social monogamy with high EPP, we performed a phylogenetic generalized least square (PGLS) analysis and a phyANOVA with the four combinations of mating system and EPP (Monogamy+Low EPP, Monogamy+High EPP, Polygyny+Low EPP, Polygyny+High EPP) as the discrete parameter and include these results in the supplement. In the PGLS, the effect of the interaction between mating system and EPP on syllable repertoire is not significant, while the effect of EPP alone is. The phyANOVA, however, points to one weakly significant association ($p=0.045$): the Polygyny+LowEPP group ($N=6$) had larger repertoires than the Monogamy+HighEPP group ($N=31$). The species exhibiting polygyny with low EPP had the highest syllable repertoire, though this was also the group with the fewest species. This could be evidence that points toward EPP modulating the male fitness stratification under polygynous mating systems. This is an interesting question that should be investigated further with more species data as it becomes available in the future. We have added text regarding these tests and conclusions to the Methods, Results, and Discussion.

Abstract:

The abstract gives short shrift to the results of the analyses with respect to EPP. If the central goal of the study is to test the effects of sexual selection (as measured by reproductive variance in males) on song evolution, then EPP should at least be mentioned.

Thank you. We have added text to the first part of the abstract to place more focus on EPP.

Introduction:

The two concepts of polygyny and EPP are introduced separately (in the first and fourth paragraph, respectively), sandwiching a discussion of song complexity. This is in part due to the structuring of the first paragraph around Darwin. While this works for emphasizing the importance of mating systems (polygyny in particular) in evolutionary theory, it does not provide room for discussing EPP (because Darwin didn't know about it). Thus, I wonder if this structure does the study disservice by not providing the logical connection between testing both polygyny and EPP in this study.

Thank you; we agree that the framing around Darwin does not serve the full scope of the paper. We have restructured the introduction replacing the Darwin reference with more consideration of EPP.

Line 55-57. This sentence starting with "Song in oscine songbirds could be a particularly salient target for sexual selection...": This statement is correct and interesting, but seemed out of place as the end to the paragraph specifically about syllable repertoire size.

We moved and edited this sentence, which we think has improved the flow of the text.

Lines 185-203: While I understand the reasoning for including these nonparametric tests (i.e., to illustrate what the results might look like using phylogenetically uninformed methods), I think it perhaps takes the reader on a detour that detracts from the main findings based on a phylogenetic methods. One compromise I would propose is to utilize the fact that Figure 1 already visually illustrates what the results look like without phylogenetic controls. So perhaps the authors could simply add a sentence in the previous paragraph (presenting results of the first row in Figure 1) pointing out that this figure shows how, even when some metrics of song complexity appear to have higher values in polygynous mating systems, phylogenetically controlled tests show that mating system has no effect. This would could then flow straight to the results with respect to EPP without detouring to the “what the results would’ve looked like if we had used poor stats”.

We appreciate and agree with the feedback and have removed the parts of the analysis which do not include phylogenetic control from the main text, supplement, and figures.

Lines 164-166: This caveat about the Jetz tree can probably be removed without sacrificing clarity. I’m not sure that anyone will fault the authors for not using a global phylogeny of birds that are not yet based on full genomes at this point.

Thank you. We have removed this sentence.

Lines 173-175: I think this sentence could be removed without sacrificing clarity—I think the point that controlling for phylogeny is important is sufficiently clear already.

Agreed; we have removed this sentence.

Line 268-274. These two sentences flip back and forth from saying the null hypothesis was tested, then saying the alternative was assessed. Perhaps it could be shortened to one sentence, e.g.: “We then used the Brownie algorithm to test whether mating system affects evolutionary rates by comparing the average log-likelihood of the one-rate model (i.e., null model: evolution at same rate regardless of mating system) to that of the two-rate model (i.e., rate of evolution differs in monogamous vs. polygynous states: Supplementary Figs. 14–15).”

Thank you; we have rephrased this passage.

Lines 448-451: While it is always true that there are multiple potential evolutionary dynamics. However, it would just be simpler to state the evolutionary dynamic that you detect. Thus, you could delete this sentence and start the paragraph with the next sentence.

We agree and have removed this sentence.

Lines 483-505: This paragraph is filled with speculations, which is not very strong. I wonder if it would be space better occupied by a paragraph reconciling the difference between polygyny and EPP, as mentioned in one of my first comments.

Thank you for this perspective. We trimmed down this paragraph and shifted the focus more towards “ongoing questions” rather than speculations.

Lines 507-523: This paragraph is somewhat repetitive with the paragraph following it, as they both deal with limitations of the approach for interpretation. I think you could delete this paragraph, and then incorporate a couple of points in the following paragraph (e.g., one sentence about how including/excluding Mimidae seems to have an outsized effect).

Lines 529-531: This sentence is repetitive from paragraph above it.

We agree, and we have merged these two paragraphs into one more concise paragraph that includes the results of removing the Mimidae from our analysis.

Figures 5 & 6: I think I understand this figure, but I am a bit confused about the arrow sizes. The width of the line and the size of the arrowhead seems decoupled, with some arrows having a larger head than arrows associated with larger values (e.g., Figure 5b, arrow from poly/medium to poly/small has larger head than arrow from poly/small to poly/medium, despite having smaller transition value). Does this signify something, or is it just a coding mistake?

We apologize for the confusion; we manually edited some of the arrowheads so that they would not overlap with the text as the line width increased, and we edited some of them inconsistently. We have edited these figures to standardize the arrowheads so that they scale with the line width.

Reviewer #2 (Remarks to the Author):

This is a well-written, thorough investigation into two main hypotheses thought to drive the evolution of elaborate birdsong - a very complex trait nearly ubiquitous throughout songbirds. The authors create and use a large, comprehensive data set to evaluate their predictions, in one of the most direct assessments of this question that I have seen for some time. Their results are counter to expectations that song is directionally selected in systems in which sexual selection is expected to be high. The methods are robust.

We appreciate your detailed reading of the manuscript and your helpful suggestions!

I have a number of logistical issues:

1. The authors should de-emphasize their focus on the non-phylogenetically controlled analyses - as this is inappropriate practice.

We agree with this feedback and have removed the parts of the analysis which do not include phylogenetic control from the main text and supplement.

2. At each section in which the song metrics are discussed, it is never clear to me where these data come from and it is not clear until the methods how they are classified. Some mention of this sooner and more detail in explaining compilation of these metrics is important, since this is the focus of the paper.

Thank you; we now explain the sources and definitions of the song data much earlier in the paper. In the Introduction, we now specify that we gathered the song metrics from published literature and curated field guides. We list all of the song metrics in Table 1 in the introduction, so in the caption of this table we now give descriptions of each song term and mating system term used in this study.

3. The authors taut a large data set, but because they only have song data for ~350 species, this limits the analysis to this smaller sample size (which is still good for this hard-to-come-by song data).

We have tried to be very explicit about our sample size for each data type, and we report and make publicly available all mating system and song data that we collected, not just the data for the species that overlap between the data types, so that the database might be useful to other researchers studying mating system or song in other contexts.

4. There is very little discussion of alternative functions of song or trade-offs with other selections pressures that may be at play. It would be nice to have a paragraph in the discussion dedicated to explanations surrounding these obvious reasons for why the predicted trend was not found (especially given high phylogenetic signal).

We include a much-revised paragraph in the discussion that more clearly discusses the other selection pressures that might be at play for different species, including territory quality, plumage, etc. We also mention the high phylogenetic signal of the song metrics in the discussion.

5. Discussion, overall could be shorter - getting right to explanation for why the authors believe they see these results.

The discussion has been substantially trimmed, hopefully providing a more succinct explanation of our results.

Specific comments follow:

Line 19: the authors state here that their analysis includes the full songbird phylogeny, however, their data on mating system, song, and EPP only exists for a portion of the songbirds.

Thank you for pointing this out. We see how this was unclear and changed wording to “alongside songbird genetic phylogenies”.

Line 20: syllable repertoire size classified how?

We have edited the abstract, which now includes a brief explanation of how we defined syllable repertoire size.

End of abstract: Why do you think these two contrary results occur? What implications does this have for how we think about and study bird song?

We now explore these issues more thoroughly in the discussion, but since there is no simple answer to these questions and since the abstract is limited to 150 words, we could not do justice to these important questions in the abstract itself.

Line 46: “courtship song” assumes that song in songbirds is used exclusively for attracting mates and/or that you focused exclusively on courtship song.

Thank you for bringing this to our attention. We removed “courtship” and now briefly list functions of song.

However, this would be difficult, as song likely serves multiple functions in a lot of species (especially non-polygynous, year-round territorial species) and you may not know the function a priori to sampling. In fact, the multi-functionality of song could be a reason for your unexpected results. Try to avoid specifying concretely the function of song prior to analysis. Also, discussing some of these alternative functions (e.g., territory defense and mediating social interactions) could be worth discussing in intro to refer to later.

Thank you for raising this point. We have broadened the listed functions of song to read: “and has functions in mate choice, intrasexual competition, and mediating other social interactions,” which avoids specifying functions concretely.

Line 121: use the term “shared ancestral history” throughout. Can state “shared ancestry” or “phylogeny” instead.

Thank you for this suggestion. We have streamlined the terminology that we use to discuss the phylogenetically controlled analyses in the text.

Line 149: please explain a bit about where these songs came from, any structure to the data (multiple individuals, geographic locations, subspecies per species, etc). How did you select the species you included?

We added “from published literature and curated field guides” to this sentence and a note to see the Methods for the full data curation protocol.

Lines 149–150: please explain how these song metrics (syllable repertoire size, song repertoire size, song interval (what is this?), and song duration) were calculated/quantified. If these values came from the literature, how was it ensured that these metrics had approximately similar meaning/were assessed similarly among species? Differences in authors’ interpretation or calculation of repertoire sizes could lead to different estimates.

The song metrics have now been defined in the Introduction (in description under Table 1) instead of just in the Methods. The nitty-gritty details of how we assembled the song database are presented in the methods, but we now mention the issue of differences in defining song features in the Table 1 description in the introduction and note that details are provided in the methods.

Line 152-153: How did you arrive at this sample size? Above you state you have song data for 352 oscines. Below you state you have data for 39 suboscines. How does this total 899? The function of this data set seems limited by the species with song data, as you need the song data to conduct the comparative analyses. So, even if you have more species with mating system or EPP data, the total number of species in the analysis should be dictated by those with song metrics. 300+ species with repertoire data is still impressive.

We have rephrased this sentence to clarify that 890 is the total number of species with one or more datapoint in the database. This number includes species for which we only have mating system data.

While these species are not directly represented in all tests, they are used in the calculation of rates of evolution of mating systems, which is then used in subsequent tests that only include those species for which there is also song data. Even if we couldn't include a species in the analysis, we opted to leave it in the database, which we are making publicly available for researchers who might find the full dataset useful in different contexts (i.e. studying mating system outside of the context of song).

Note: We revised the number of species from 899 to 890, because 9 species had tentative data that we gathered but subsequently eliminated prior to completing any analyses. The tentative data for the 9 species is included in the notes in Supplementary Data 1, but the values listed are "NA" and the information was not analyzed.

Please state how many species were included in the comparison to mating system and EPP, respectively.

For each comparison between mating system/EPP and a song feature, the total number of species included in that comparison is now clearly stated on the associated boxplot in Figure 1 and in Table 3.

Line 165: Only species with genetic data in the Jetz tree should be used for certain kinds of comparative analyses (especially those involving rate), which you can specify when downloading trees. Did you take this into consideration?

Thank you for this suggestion. All species used in our analyses involving Syllable Repertoire have genetic data in the Jetz tree. Some species in the other song trait analyses did not, but none of the analyses using these trees indicated significant differences in evolutionary rate. We have repeated all of our analyses with trees downloaded with the specification that only species with genetic data are included. These analyses did not lead to any different conclusions from our original analyses. We include the results from all of these analyses in the supplemental information and discuss them in the main text.

Line 170-171: this is an odd way to state this. The controlling for phylogeny controls for the statistical non-independence that potentially exists because some species are more closely related than others. Any more, this is standard practice and does not require much explanation.

Thank you. We have removed this sentence.

Line 174-175: a number of studies have shown that it is important to control for phylogeny whether phylogenetic signal is high or not (because of the need to control for the statistical non-independence; and that these metrics represent the signal in the residuals and not necessarily the traits themselves). To avoid promoting improper practice, I would remove this sentence.

Thank you. We have removed this sentence, as well as the methods and results which include the non-phylogenetically-controlled Wilcoxon test.

Which traits are these calculations for ?

We removed the sentence mentioned above, which also had ambiguous wording mentioned here. In the previous sentence, we now specify that all seven of the song traits had significant phylogenetic signal.

Line 180: How many anovas did you perform? One for each song feature? Did you correct for multiple comparisons?

You are correct, and we have rephrased this sentence so that it is clear that we performed one ANOVA for each song feature and corrected for multiple comparisons.

Why didn't you use pgl's or phylogenetic mixed models or other analyses that would have allowed you to simultaneously test for multiple predictors that might influence mating system or EPP concurrently? Such models (esp phylo mixed models) would also allow you control for repeated measures (inclusion of multiple samples per species or individual). Did you have any data structure like this? Again, more explain about this dataset, where it came from, and its structure are needed. Analysis: did you have any repeated measures for individuals or species? Did you control for this in your model structure?

Thank you for this suggestion. We now report a new PGLS analysis and a phylogenetic mixed model analysis using mcmcGLMM, which allowed us to account for mating system, EPP, and the interaction between the two when we analyze each song feature. We do have multiple measures for some species, and we had previously accounted for this by repeating our analysis with the minimum, median, and maximum values. Now, we streamline this process with the GLMM analyses, which incorporates all of our measurements at once. We now explicitly state this at the beginning of the results section.

Both the PGLS analysis and the GLMM analysis have very similar results: EPP is significantly associated with syllable repertoire size, which is the same association we found with the phyANOVA. There was a weak association between EPP and syllables per song with phyANOVA that was not significant after controlling for multiple hypotheses; this relationship was significant with PGLS but not with GLMM. There were no significant interactions between EPP and polygyny for any song feature. We present these results in the main text and include the relevant figures and tables in the supplemental information.

Lines 185-191: this is not standard practice (analyses should not be run to see what results would be if you did not control for phylogeny, as the analysis controlling for phylogeny is the correct statistical test). You can include this in the discussion, but I would not include it here.

Thank you. We have removed this sentence.

Lines 198-200: again, too much focus on controlling vs not controlling for phylogeny

We have shortened this section and removed the focus on phylogenetic correction.

Line 213-214: You do not discuss differences between controlling for phylogeny here (which is fine, as this should not be a major focus). If you discuss this in the above paragraph, put this statement there.

Thank you. We have removed this sentence.

Figure 1: this is the first you mention how you scored polygyny. Include in text.

Thank you, we now define our scoring method in the description of the traits in Table 1, and we note in the Introduction that definitions of the traits can be found there.

Because you should not run phylogenetically uncontrolled analyses in your situation, I would not present the Wilcoxon non-parametric results here. Alternatively, you could highlight marginally significant values, if you think it appropriate.

We have removed the phylogenetically uncontrolled analyses from the text and figures.

What is continuity? Mention in text.

Thank you -- in addition to the Methods, this is now defined in the description of the traits in Table 1, and we note in the Introduction that definitions of the traits can be found there.

Lines 264-266: did you run these analyses on multiple alternative versions of the tree ?

Thank you for this great suggestion; we now run these analyses (phylANOVA, Brownie, and BayesTraits) on multiple alternative versions of the tree. We found consistent results with this analysis compared to our analysis with the consensus tree. These results are now included in the supplemental information.

Line 427: "by extracting information from published sources"... This is the first time you mention this. Mention sooner.

We now mention in the Introduction and at the very beginning of the Results section that the database was compiled from published sources and curated field guides.

Lines 448-481: a lot of repetition here - re-explaining the findings, which were just explained in previous sections. Given short format of this journal, consider cutting or greatly reducing text here to quickly re-emphasize the main findings.

We agree and have made extensive cuts to this section to make the text more succinct and less repetitive.

Line 483: this is what the reader is waiting for. get to this important paragraph faster.

We have trimmed and rearranged the discussion so that this paragraph is earlier. We have also edited this paragraph at the request of Reviewer 1.

Methods could be organized a bit more clearly to point to sections relevant to assembly of each of the three data sets

We have restructured this portion of the methods section so that there are multiple subsections for each portion of the dataset (e.g. “Mating System and Extra-pair Paternity Data” and “Song Data”).

Line 674: to me, this appears to be the start of the section explaining song variables, but it is still not clear up front where these data came from...

We have now been explicit throughout where we sourced the data in our dataset. In addition, the reviewer is correct that this section was not very clear, so we have divided this part of the Methods into subsections so that it is explicit which data collection methods pertain to the song data and which pertain to the mating data.

Reviewers' Comments:

Reviewer #1:

Remarks to the Author:

The writing has been made concise in parts and the manuscript flows better than the original submission, and I think it enhances the impact of the work. However, somehow the manuscript is still the same length, and I can still pick out many places where redundant sentences or entire tangential paragraphs could be cut. I point some of these cases out below. Overall, however, the authors have addressed my main comments. I think the new analyses are nice complements to the main results. I just have minor, mostly editorial suggestions:

Lines 47-65: This is a very thorough explanation of how EPP could arise, and how it could affect sexual selection (either increase or decrease). But I think that

Line 90: *Acrocephalus* should be italicized

Lines 176-177: The Greek symbols did not convert in this pdf. It doesn't matter much for review purposes, but authors should make sure to double-check.

Lines 188-190: I don't think I caught this on the first reading, but are these P-values what is contained in the Read & Weary study? If so, this is confusing, because I kept looking for these P-values in Figure 1 and did not find them. I don't think it's necessary to report the p-values reported in another study, particularly because these values are pretty much meaningless without knowing what the data and statistical tests are.

Lines 266-268: This sentence is redundant with the sentence above.

Lines 311-319: I think this paragraph could be cut without sacrificing clarity for the readers. Most of this paragraph just restates what was presented in the previous section, or sets up what is explained in the subsequent paragraph.

Figures 5 & 6: I think the figure and caption got mixed up between these two figures.

Lines 321-352: The authors have segmented the syllable repertoire sizes into thirds, but I was left wondering whether one should arbitrarily segment the syllable repertoire sizes in this way. For example when looking at EPP + syllable repertoire size, the segmentation into thirds cuts through an interesting peak at repertoire size of 10. It made me wonder if one specifically looked at the relationship between transition in this range of syllable repertoire (say, 8-20), does that reveal some different dynamic about EPP and song? There is also a similar peak with Polygyny around syllable repertoire around 40. If conducted, I think results of such exploration could be reported in a supplemental section.

Lines 544-546: This sentence is unnecessary, as it just restates the method you used.

Lines 548-560: I think the beginning of this paragraph undersells your contribution. It makes it sound like what you did was mainly to add more species to existing approaches. Moreover, most of this paragraph simply restates your results, which has already been restated at the beginning of Discussion.

More generally, I think the current Discussion lacks an overarching take-home message (not just restating the results). I wonder if the manuscript may benefit by ending at Line 506 with perhaps the

addition of a concise statement about what this work says about how we should investigate the link between sexual selection and oscine song complexity in the future, or potentially, how we should study the evolution of putatively sexually selected traits in a phylogenetic framework. I think there are larger lessons than authors have stated thus far, and this is what will wrap this paper up nicely.

Signed,
Daizaburo Shizuka

Reviewer #2:

Remarks to the Author:

This manuscript is much improved from the previous version, and will make a great contribution to our understanding of how sexual selection shapes elaborate traits. I greatly appreciate the care and thoroughness with which the authors addressed reviewer comments. I have no additional comments.

Reviewer #1 (Remarks to the Author):

The writing has been made concise in parts and the manuscript flows better than the original submission, and I think it enhances the impact of the work. However, somehow the manuscript is still the same length, and I can still pick out many places where redundant sentences or entire tangential paragraphs could be cut. I point some of these cases out below. Overall, however, the authors have addressed my main comments. I think the new analyses are nice complements to the main results. I just have minor, mostly editorial suggestions:

Lines 47-65: This is a very thorough explanation of how EPP could arise, and how it could affect sexual selection (either increase or decrease). But I think that

Since the review was signed, we sent an inquiry to the reviewer to ask about the unfinished sentence in this comment. He suggested that we shorten this paragraph, and we have done so.

Line 90: *Acrocephalus* should be italicized

This has been fixed.

Lines 176-177: The Greek symbols did not convert in this pdf. It doesn't matter much for review purposes, but authors should make sure to double-check.

We hope this has been fixed by changing the font on the special characters.

Lines 188-190: I don't think I caught this on the first reading, but are these P-values what is contained in the Read & Weary study? If so, this is confusing, because I kept looking for these P-values in Figure 1 and did not find them. I don't think it's necessary to report the p-values reported in another study, particularly because these values are pretty much meaningless without knowing what the data and statistical tests are.

We agree and have removed these p-values.

Lines 266-268: This sentence is redundant with the sentence above.

We agree and have reworked this sentence.

Lines 311-319: I think this paragraph could be cut without sacrificing clarity for the readers. Most of this paragraph just restates what was presented in the previous section, or sets up what is explained in the subsequent paragraph.

We agree and have reworked this paragraph.

Figures 5 & 6: I think the figure and caption got mixed up between these two figures.

Thank you for catching this! We have renamed the figures.

Lines 321-352: The authors have segmented the syllable repertoire sizes into thirds, but I was left wondering whether one should arbitrarily segment the syllable repertoire sizes in this way. For example when looking at EPP + syllable repertoire size, the segmentation into thirds cuts through an interesting peak at repertoire size of 10. It made me wonder if one specifically looked at the relationship between transition in this range of syllable repertoire (say, 8-20), does that reveal some different dynamic about EPP and song? There is also a similar peak with Polygyny around syllable repertoire around 40. If conducted, I think results of such exploration could be reported in a supplemental section.

We conducted these analyses with different numbers of bins (2, 4, and 5 in addition to 3 reported in the main text). We now mention this in the Methods section and provide associated Supplementary Figures. The rate trends that emerged in these new analysis and visualization methods were consistent with the original conclusions.

Lines 544-546: This sentence is unnecessary, as it just restates the method you used.

We agree and have deleted this sentence.

Lines 548-560: I think the beginning of this paragraph undersells your contribution. It makes it sound like what you did was mainly to add more species to existing approaches. Moreover, most of this paragraph simply restates your results, which has already been restated at the beginning of Discussion.

We have revised the beginning of this paragraph to better present our main findings. We have also revised the rest of the paragraph extensively so that it is no longer a restatement of results.

More generally, I think the current Discussion lacks an overarching take-home message (not just restating the results). I wonder if the manuscript may benefit by ending at Line 506 with perhaps the addition of a concise statement about what this work says about how we should investigate the link between sexual selection and oscine song complexity in the future, or potentially, how we should study the evolution of putatively sexually selected traits in a phylogenetic framework. I think there are larger lessons than authors have stated thus far, and this is what will wrap this paper up nicely.

Some of this text was requested by the other reviewer and has been retained, but we revised this section to contextualize our work more concisely.